# Dynamic View Synthesis as an Inverse Problem

**Hidir Yesiltepe**   **Pinar Yanardag**

hidir@vt.edu   pinary@vt.edu

Virginia Tech

`https://inverse-dvs.github.io/`

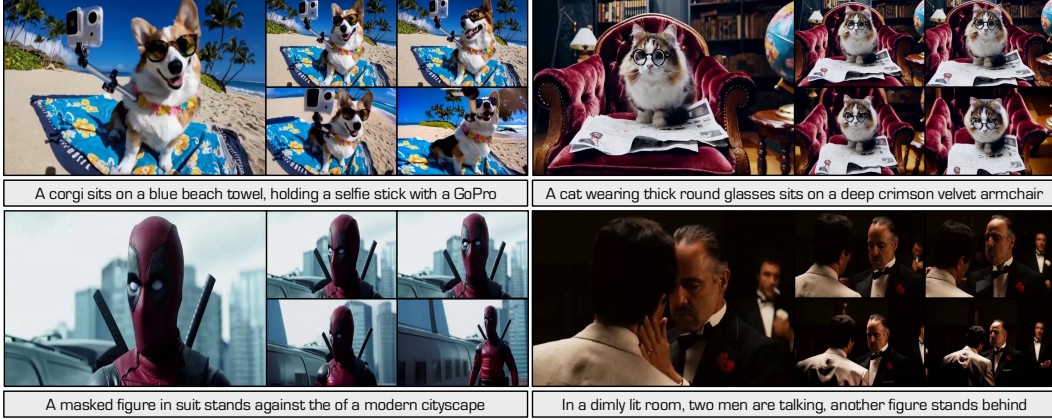

Figure 1: From real-world complex scenes to AI-generated videos, our method preserves identity fidelity and synthesizes plausible novel views by operating entirely in noise initialization phase.

## Abstract

In this work, we address dynamic view synthesis from monocular videos as an inverse problem in a training-free setting. By redesigning the noise initialization phase of a pre-trained video diffusion model, we enable high-fidelity dynamic view synthesis without any weight updates or auxiliary modules. We begin by identifying a fundamental obstacle to deterministic inversion arising from zero-terminal signal-to-noise ratio (SNR) schedules and resolve it by introducing a novel noise representation, termed K-order Recursive Noise Representation. We derive a closed form expression for this representation, enabling precise and efficient alignment between the VAE-encoded and the DDIM inverted latents. To synthesize newly visible regions resulting from camera motion, we introduce Stochastic Latent Modulation, which performs visibility aware sampling over the latent space to complete occluded regions. Comprehensive experiments demonstrate that dynamic view synthesis can be effectively performed through structured latent manipulation in the noise initialization phase.

## 1 Introduction

Dynamic view synthesis (DVS) [15, 40, 47, 57, 66] from monocular videos [12, 13, 30, 67, 52, 2, 17, 65, 59] is a computer vision task that aims to generate new, dynamic perspectives of a scene using only a single video as input. This process involves predicting how a scene would appear from angles not captured in the original footage, requiring the inference of depth, occluded regions, and unseen details. In the film industry, DVS can revolutionize post-production by enabling virtual camera sweeps through a scene or producing additional shots from different angles, eliminating the need for expensive reshoots. In robotics, it supports advanced perception systems by generating synthetic

39th Conference on Neural Information Processing Systems (NeurIPS 2025).

viewpoints that train algorithms for navigation, manipulation, and active perception tasks in complex environments.

Historically, DVS has relied on explicit 3D reconstruction methods, such as Neural Radiance Fields (NeRF) [39], its dynamic extension D-NeRF [44], K-Planes [11], and 3D/4D Gaussian Splatting [25, 56]. These seminal volumetric and point-based approaches model scenes as continuous representations or point configurations. However, they impose strict prerequisites: multi-view supervision, computationally intensive per-scene optimization, and precise camera calibration. Recently, a paradigm shift has emerged, leveraging video-diffusion [5, 61, 53, 4, 33] priors to address these limitations. Diffusion models appeal for DVS because they implicitly capture geometry [64, 38] and appearance [23, 63, 8, 62] in their latent space, inherently provide temporal consistency, and bypass the need for explicit 3D modeling. Under this diffusion paradigm, two dominant approaches have surfaced. The first involves attention-sharing architectures, as seen in Generative Camera Dolly [52], TrajectoryAttention [59], TrajectoryCrafter [65], and ReCamMaster [2]. These methods integrate camera-aware branches such as pixel-trajectory attention, dual streams, or 3D attention layers to enable fine-grained camera control. However, they require additional architectural modules and extensive retraining on large synthetic datasets like Unreal Engine 5 [10] or Kubric [16], leading to domain-gap issues when applied to natural settings. The second recipe employs LoRA-based fine-tuning, exemplified by ReCapture [67] and Reangle-A-Video [22]. These approaches attach spatial and temporal Low-Rank Adaptations (LoRAs) [20, 7, 46, 9, 68], and perform per-video fine-tuning leveraging masked losses. Across both strategies, shared limitations persist: they require updating backbone parameters or adding layers, depend on curated synthetic data or video-specific fine-tuning, and suffer from pitfalls when the inversion process misaligns with the model's forward noise schedule. These constraints underscore a critical open question: **Can we achieve 6-DoF monocular DVS without any weight updates, auxiliary modules, or synthetic pre-training purely by manipulating the initial noise fed into a video-diffusion model?**

In this work, we pioneer a fundamentally different approach to DVS from monocular videos. We demonstrate that by solely manipulating the initial noise fed into a video diffusion model, we can achieve state-of-the-art performance without any weight updates or auxiliary modules. This novel perspective shifts the focus from architectural redesign or resource-intensive retraining to efficient noise design, distinguishing our method from existing approaches. Our approach is centered around two key innovations. First, we identify and formalize the **Zero-Terminal SNR Collapse Problem**, which arises when training schedules enforce zero signal-to-noise ratio at the terminal timestep, causing a collapse in information content and obstructing deterministic inversion. To resolve this, we propose the **K-order Recursive Noise Representation (K-RNR)**, which recursively refines the initial noise in alignment with the model's forward schedule, enabling stable and faithful reconstruction of the original scene. We derive closed-form expressions for this refinement process and stabilize generation with an adaptive variant that prevents scale explosion. Second, to address the synthesis of newly visible content due to camera motion, we introduce **Stochastic Latent Modulation**, a visibility-aware sampling mechanism that directly completes occluded latent regions using context-aware latent permutations. This enables plausible scene completion in the noise initialization phase. Together, these components form a unified framework that achieves high-fidelity reconstruction and physically consistent view synthesis from monocular input. Our contributions can be summarized as follows:

- We identify and formalize the Zero-Terminal SNR Collapse Problem, showing that while zero terminal SNR schedules improve generation quality, they inherently break injectivity, preventing deterministic inversion and hindering faithful reconstruction.

- We propose K-order Recursive Noise Representation (K-RNR) to resolve the obstruction caused by the Zero-Terminal SNR Problem. By defining a recursive refinement relation between the VAE-encoded latent and the positive-SNR DDIM-inverted latent, we derive closed-form noise expression, enabling high-fidelity reconstructions of original scenes.

- We introduce Stochastic Latent Modulation (SLM), a novel latent-space completion mechanism that infers content for newly visible regions by performing visibility-aware sampling and contextual latent permutation, enabling physically plausible synthesis in occluded areas without modifying the model.

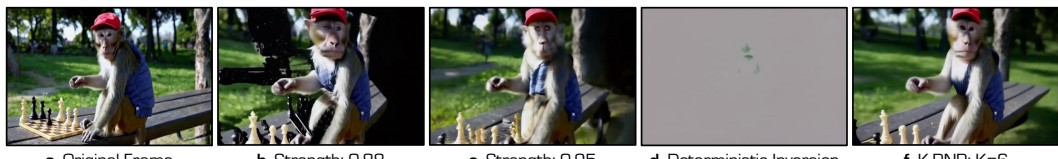

| a. Original Frame | b. Strength: 0.88 | c. Strength: 0.95 | d. Deterministic Inversion | f. K-RNR: K=6 |

Figure 2: **Approaches to Zero-Terminal SNR Collapse Problem.** Figures b-f shows the inpainted versions of Figure a. (b) Low strength preserves source content but renders unseen regions as black. (c) High strength improves propagation into unseen areas but causes identity drift. (d) DDIM inverted latent as initial noise leads to *washed-out*, high saturation generation. (f) Our K-RNR ($k = 6$) with Stochastic Latent Modulation preserves identity and completes newly visible regions with plausible content.

## 2 Related Work

This section reviews prior research in two closely related areas relevant to our work. The first is novel view synthesis for dynamic scenes, and the second is video-to-video translation with camera control.

**Novel View Synthesis for Dynamic Scenes.** Novel view synthesis seeks to generate unseen perspectives from available visual data, with substantial advancements driven by neural rendering. For static scenes, Neural Radiance Fields (NeRF) [39] and 3D Gaussian Splatting [25] provide detailed 3D reconstructions. Dynamic scene extensions, such as D-NeRF [44], K-Planes [11], HexPlane [6], and HyperReel [1], depend on synchronized multi-view inputs, which are often impractical for casual settings. Monocular video methods, including Neural Scene Flow Fields [31], DynIBaR [32], Robust Dynamic Radiance Fields [36], and Dynamic View Synthesis [12], utilize depth-based warping or neural encodings but face challenges with occlusions and extrapolation beyond input views. Recent approaches, such as 4D Gaussian Splatting [56], Dynamic Gaussian Marbles [50], and GaussianFlow [14], enhance efficiency with 3D Gaussian representations, yet require robust multi-view data or significant input camera motion, restricting broader applicability.

**Video-to-Video Translation with Camera Control.** Early video-to-video translation efforts, such as World Consistent Video to Video [37] and Few Shot Video to Video [55], targeted tasks like outpainting. Generative Camera Dolly [52] trains on synthetic multiview videos from Kubric, but domain gaps limit generalizability in natural settings. ReCapture [67] uses a two stage pipeline that first generates an anchor video with CAT3D [15] multiview diffusion or point cloud rendering, followed by refinement using spatial and temporal LoRA modules. However, per video optimization hampers scalability. Methods like DaS [17] and GS DiT [3] enforce 4D consistency through 3D point tracking with tools such as SpatialTracker [58] and Cotracker [24], though tracking inaccuracies in complex scenes limit effectiveness. ReCamMaster [2] proposes generative rerendering within pre-trained text to video models using with a frame-conditioning attention sharing mechanism using a large Unreal Engine 5 [10] dataset, but struggles with high computational cost as the number of tokens are doubled in the 3D attention mechanism. TrajectoryCrafter [65] decouples view transformation and content generation using a dual stream diffusion model conditioned on point clouds and source videos, but remains constrained under large camera shifts. Trajectory Attention [59] applies pixel trajectory attention for camera motion control and long range consistency, however, it is sensitive to sparse or fast motions and lacks full 3D consistency.

## 3 Background

In this section, we review the base video diffusion model in §3.1, followed by common noise initialization strategies used in current video models for I2V and V2V applications in §3.2.

### 3.1 Base Video Diffusion Model

Following prior works [65, 17], our work builds upon the I2V variant of the CogVideoX [61]. CogVideoX is a transformer-based video diffusion model operating in latent space with a $4\times$ temporal and $8\times$ spatial compression. The model takes a single RGB image $\mathbf{I} \in \mathbb{R}^{H\times W\times 3}$ as input

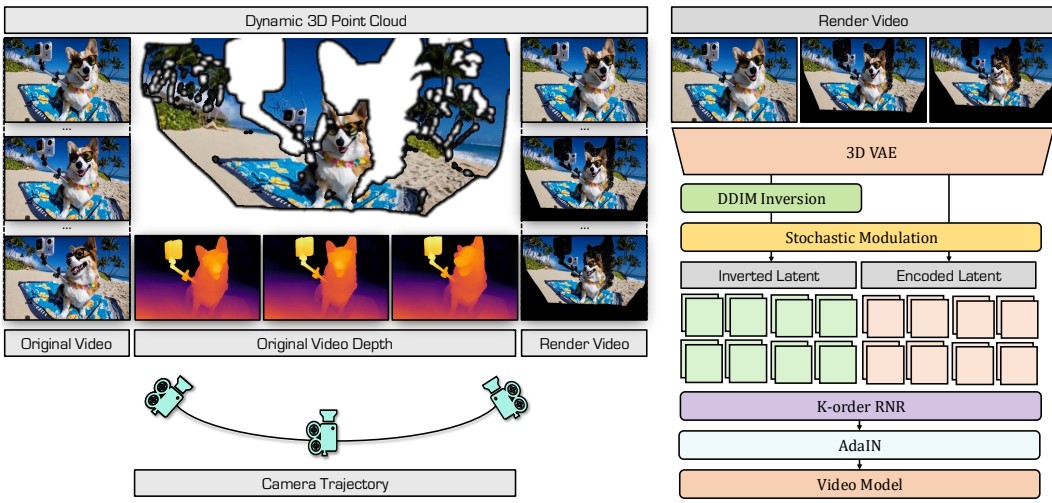

Figure 3: **Overview of Our Method.** (Left) We lift a monocular video into a dynamic 3D point cloud and render novel views under target camera trajectories, revealing unseen regions. (Right) Our method synthesizes coherent outputs by initializing noise with DDIM inversion, Stochastic Latent Modulation, and K-order Recursive Noise Representation, without modifying the video model.

and generates a video $\mathbf{V} \in \mathbb{R}^{\mathrm{F \times H \times W \times 3}}$ with F frames. The image is first encoded by a 3D VAE [26] into a spatial latent $\mathbf{z}_{\mathrm{img}}$ of size $\mathrm{C} \times \frac{\mathrm{H}}{8} \times \frac{\mathrm{W}}{8}$, with $\mathrm{C} = 16$. To extend this representation across time, it is broadcast along the temporal dimension and concatenated with $\frac{\mathrm{F}}{4} - 1$ zero latents, forming a tensor $\mathbf{x}_0$ of size $\frac{\mathrm{F}}{4} \times \mathrm{C} \times \frac{\mathrm{H}}{8} \times \frac{\mathrm{W}}{8}$. Finally, $\mathbf{x}_0$ is concatenated with a noise tensor $\epsilon \sim \mathcal{N}(0, \mathbf{I})$ along the channel dimension, yielding the initial noisy input $\mathbf{x}_{\mathrm{init}}$ of size $1 + \left\lceil \frac{\mathrm{F\text{-}1}}{4} \right\rceil \times 2\mathrm{C} \times \frac{\mathrm{H}}{8} \times \frac{\mathrm{W}}{8}$ for the I2V task.

## 3.2 Noise Initialization Strategies

Current video generation models [54, 27, 53, 61] for Image-to-Video (I2V) and Video-to-Video (V2V) tasks typically employ specific noise initialization strategies. These strategies can be broadly categorized into two main groups: **deterministic inversion** and **schedule-consistent interpolation**.

**Deterministic Inversion.** In models such as ModelScope [54], the network is conditioned on a discrete sequence of timesteps $\{t = 0, \ldots, T\}$, with each timestep associated with a strictly positive cumulative signal coefficient $\bar{\alpha}_t > 0$. In this setting, the clean latent representation can be deterministically mapped to the noise manifold using DDIM Inversion [49].

**Schedule-consistent Interpolation.** In contrast, standard DDIM inversion is not directly applicable when the network is conditioned on a continuous sequence of timesteps, as in models like SVD [4]. In such cases, the initial noisy latent is initialized as $\mathbf{x}_{\mathrm{init}} = \mathbf{x}_0 + \gamma \cdot \epsilon$, where $\gamma$ is a noise augmentation parameter that controls the strength of the initial image perturbation. In the Flow Matching-based [35] video model HunyuanVideo [27], the initial noisy latent at a discrete timestep $t \in \{0, \ldots, T\}$ is given by $\mathbf{x}_{\mathrm{init}} = t \cdot \epsilon + (1 - t) \cdot \mathbf{x}_0$ for I2V applications. Similarly, in Wan [53], another Flow Matching model, the noise initialization is defined as $\mathbf{x}_t = \sigma_t \cdot \epsilon + (1 - \sigma_t) \cdot \mathbf{x}_0$, where $\sigma_t$ is a schedule-dependent weighting factor. CogVideoX [61] is trained with zero terminal signal-to-noise ratio (SNR), which makes DDIM inversion not directly applicable as we discuss in §4.1. In V2V translation tasks, it initializes the noisy latent as $\mathbf{x}_{\mathrm{init}} = \sqrt{\bar{\alpha}_t}\mathbf{x}_0 + \sqrt{1 - \bar{\alpha}_t}\epsilon$ with signal-to-noise-ratio $\mathrm{SNR}(t) = \frac{\bar{a}_t}{1 - \bar{a}_t}$.

## 4 Methodology

Dynamic view synthesis involves simultaneously (1) preserving scene fidelity and (2) completing newly visible regions as the camera moves. This requires not only faithfully reconstructing identities and actions over time but also plausibly synthesizing previously unseen regions. To address the former, we first define the **zero terminal SNR collapse** problem, which reveals the incompatibility between deterministic inversion and schedule-consistent interpolation in models like CogVideoX trained with zero terminal SNR (§4.1). We resolve this with **K-order Recursive Noise Representation** , which

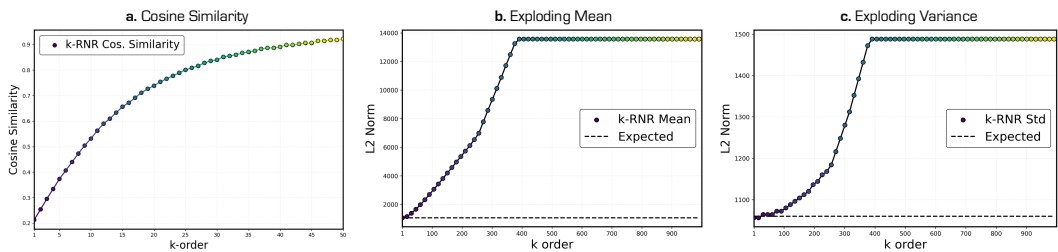

Figure 4: **K-RNR Analysis** (a) Cosine similarity between $\epsilon^{(k)}$ and VAE-encoded latent $\mathbf{x}_0$. (b) For increasing $k$ values, the mean and (c) the variance of $\epsilon^{(k)}$ explodes.

enables effective use of DDIM-inverted latents in such settings (§4.2). To address the latter, we propose a stochastic latent modulation strategy that infers unseen regions resulting from camera motion (§4.4).

## 4.1 Zero Terminal SNR Collapse

We begin our discussion by identifying a key issue that hinders the direct use of DDIM-inverted latents during the noise initialization phase under zero-terminal SNR noise schedules. Lin et al. [34] argue that noise schedules should enforce zero SNR at the final timestep and that sampling should always start from $t = T$ to ensure alignment between diffusion training and inference. Based on this principle, CogVideoX [61] adopts a zero terminal SNR during training following the noise schedule used in [45]. While this setup improves generation quality and ensures consistency between training and inference, we show that it causes a breakdown in injectivity.

**Proposition 4.1.** Let $\{\alpha_t\}_{t=0}^T$ be a variance-preserving noise schedule with cumulative products $\bar{\alpha}_t = \prod_{s=1}^t (1 - \beta_s)$, such that the schedule enforces zero terminal SNR with $\bar{\alpha}_T = 0$. Define the forward diffusion map

$$\Phi_T(x_0, \epsilon) = \sqrt{\bar{\alpha}_T} x_0 + \sqrt{1 - \bar{\alpha}_T}\, \epsilon, \quad \epsilon \sim \mathcal{N}(0, I).$$

Then, for every pair of latents $x_0, x_0' \in \mathbb{R}^d$ and every noise sample $\epsilon$,

$$\Phi_T(x_0, \epsilon) = \Phi_T(x_0', \epsilon) = \epsilon.$$

Hence $\Phi_T(\cdot, \epsilon)$ is not injective in $x_0$. Consequently, deterministic inversion methods such as DDIM inversion cannot uniquely recover $x_0$ from $x_T$.

Proposition 4.1 implies that a noise schedule with zero terminal SNR forces the schedule-consistent latent at the last time step to be

$$\mathbf{x}_T = \sqrt{\bar{\alpha}_T}\, \mathbf{x}_0 + \sqrt{1 - \bar{\alpha}_T}\, \boldsymbol{\epsilon} = \boldsymbol{\epsilon}, \qquad \boldsymbol{\epsilon} \sim \mathcal{N}(0, \mathbf{I}),$$

which collapses to pure noise because $\bar{\alpha}_T = 0$. No information from the original frame $\mathbf{x}_0$ survives, so the resulting video-to-video translation cannot remain aligned with the source content. A common workaround is to begin sampling from an earlier index $t < T$ for which $\bar{\alpha}_t > 0$. However, it shortens the diffusion trajectory and therefore limits translation diversity, which is an important component of dynamic view synthesis. As shown in Fig.2(b), this also results in the reconstruction of regions that are unseen after camera transformation. Even when $\bar{\alpha}_t$ is very small but non-zero, the stochastic term $\boldsymbol{\epsilon}$ introduces perturbations that accumulate during generation and ultimately lead to identity drift as demonstrated in Fig.2(c).

## 4.2 K-order Recursive Noise Representation (K-RNR)

An alternative workaround to the zero-terminal SNR collapse problem is to perform DDIM inversion with a positive terminal SNR, allowing the resulting latent to initialize the diffusion process for downstream tasks. However, as shown in Fig.2(d), this approach still results in images with a washed-out appearance. We attribute this issue to a mismatch between the scale of the expected initial noise and that

Figure 5: **Expected Norm Deviation**

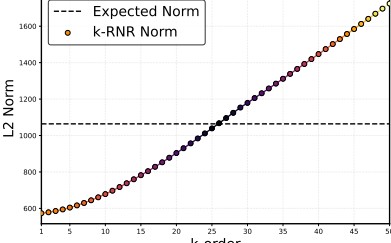

produced by schedule-consistent interpolation, given by $\mathbf{x}_{\text{init}} = \sqrt{\bar{\alpha}_t}\,\mathbf{x}_0 + \sqrt{1 - \bar{\alpha}_t}\,\boldsymbol{\epsilon}^{\text{inv}}$, evaluated at $t = 0.95T$. This discrepancy is visualized in Fig.5 along the $k = 1$ axis. Moreover, applying normalization or standardization to the $\mathbf{x}_{\text{init}}$ introduces trajectory drift, leading to degraded results, as demonstrated in Supplementary Material.

Given the limitations of existing workarounds for the zero-terminal SNR collapse problem, we propose a new noise initialization mechanism, **K-order Recursive Noise Representation** , which aligns deterministic inversion with schedule-consistent interpolation. In this formulation, we treat the VAE-encoded latent $\mathbf{x}_0$ as the *pivot latent*, and define the initial noise as $\mathbf{x}_{\text{init}} = \boldsymbol{\epsilon}^{(k)}$. Throughout the paper, we use superscripts enclosed in parentheses to denote recursion order, while superscripts without parentheses indicate exponentiation.

**Proposition 4.2.** Let $\mathbf{x}_0 \in \mathbb{R}^d$ be the pivot latent and let $\bar{\alpha}_t > 0$ denote the cumulative signal coefficient at timestep $t$. Define the recursive noise initialization by:

$$\boldsymbol{\epsilon}^{(1)} = \sqrt{\bar{\alpha}_t}\,\mathbf{x}_0 + \sqrt{1 - \bar{\alpha}_t}\,\boldsymbol{\epsilon}^{\text{inv}},$$

and for $k > 1$,

$$\boldsymbol{\epsilon}^{(k)} = \sqrt{\bar{\alpha}_t}\,\mathbf{x}_0 + \sqrt{1 - \bar{\alpha}_t}\,\boldsymbol{\epsilon}^{(k-1)}.$$

Then, for a discrete recursion depth $k \in \mathbb{N}_{>0}$, the closed-form expression for $\boldsymbol{\epsilon}^{(k)}$ is:

$$\boldsymbol{\epsilon}^{(k)} = \left( \sum_{i=1}^{k} \sqrt{\bar{\alpha}_t}\, \left(\sqrt{1 - \bar{\alpha}_t}\right)^{i-1} \right) \mathbf{x}_0 + \left(\sqrt{1 - \bar{\alpha}_t}\right)^{k} \boldsymbol{\epsilon}^{\text{inv}}. \tag{1}$$

Which can be generalized to continuous recursion depth $k \in \mathbb{R}_{>0}$ as:

$$\boldsymbol{\epsilon}^{(k)} = \left( \sqrt{\bar{\alpha}_t}\, \frac{1 - \left(\sqrt{1 - \bar{\alpha}_t}\right)^{k}}{1 - \sqrt{1 - \bar{\alpha}_t}} \right) \mathbf{x}_0 + \left(\sqrt{1 - \bar{\alpha}_t}\right)^{k} \boldsymbol{\epsilon}^{\text{inv}}. \tag{2}$$

Refer to the Appendix for the proofs of Eq.(1-2). By treating $\mathbf{x}_0$ as a pivot latent and recursively updating the noise latent $\boldsymbol{\epsilon}^{(i)}$, the resulting initialization $\mathbf{x}_{\text{init}} = \boldsymbol{\epsilon}^{(k)}$ becomes increasingly aligned with the structure of $\mathbf{x}_0$. We quantify the alignment by measuring the cosine similarity between $\mathbf{x}_0$ and $\boldsymbol{\epsilon}^{(k)}$, as shown in Fig.4(a). To isolate the effect of the inverted latent $\boldsymbol{\epsilon}^{\text{inv}}$, we initialize the recursion with $\boldsymbol{\epsilon}^{(1)} = \sqrt{\bar{\alpha}_t}\,\mathbf{x}_0 + \sqrt{1 - \bar{\alpha}_t}\,\boldsymbol{\epsilon}$, where $\boldsymbol{\epsilon} \sim \mathcal{N}(0, \mathbf{I})$, and apply the recursive formulation with discrete depth as described in Proposition 4.2. As $k$ increases, the similarity steadily improves, indicating that K-RNR progressively enhances structural fidelity by injecting more of the original latent structure into the initialized noise.

Importantly, this growing similarity is not the only factor contributing to improved reconstruction quality. As shown in Fig.5, the expected scale of $\boldsymbol{\epsilon}^{(k)}$ also becomes better aligned with the reference distribution scale as $k$ increases, up to a certain threshold. This alignment is achieved without applying explicit normalization or standardization which results in high saturation generations.

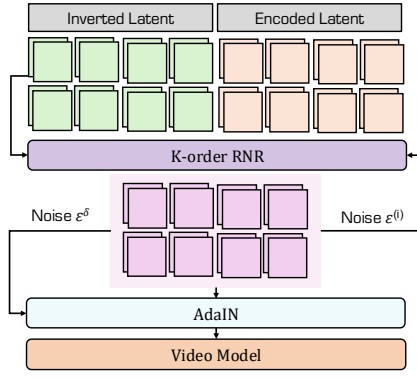

Figure 6: **Adaptive K-RNR**

However, K-RNR on its own suffers from exploding mean and variance, as demonstrated in Fig.4(b–c). This indicates that as the recursion order $k$ increases, the scale of the initialized noise grows rapidly. In practice this problem leads to high contrast outputs with exploded RGB colors in the generated video. To address this issue, we introduce **Adaptive K-RNR**, which stabilizes the recursion by incorporating scale information from an intermediate recursion step. Specifically, given a total recursion depth $k$, we select an intermediate index $\delta \in \{1, \ldots, k\}$, compute the intermediate noise representation $\boldsymbol{\epsilon}^{(\delta)}$, and apply $\tilde{\mathbf{x}}_{\text{init}} = \text{AdaIN}\left[\boldsymbol{\epsilon}^{(k)}, \boldsymbol{\epsilon}^{(\delta)}\right]$. This operation preserves the structural benefits of K-RNR while suppressing the scale explosion that leads to visual artifacts.

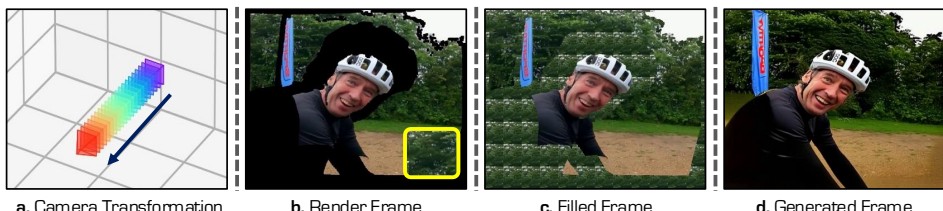

| **a.** Camera Transformation | **b.** Render Frame | **c.** Filled Frame | **d.** Generated Frame |

Figure 7: **Stochastic Latent Modulation Motivation.** To evaluate the model's capacity for physical plausibility in unseen regions, we modify the rendered input with occlusion-filling strategies. (a) Camera motion trajectory. (b) Original render frame. (c) Occluded regions are filled by repeating a background patch. (d) Resulting frame generated by combining the filled render with $\epsilon_{\text{inv}}$ using K-RNR, demonstrating plausible yet artifact-prone content synthesis in unseen areas.

## 4.3 Conditioning on Camera Information

Following prior works [65, 67, 22, 66, 59, 17], we incorporate explicit camera conditioning [29] into our framework to enable precise control over novel view synthesis. Given a source video $\mathbf{V} = \{\mathbf{I}_i\}_{i=1}^n$, where each frame $\mathbf{I}_i \in \mathbb{R}^{\text{C} \times \text{H} \times \text{W}}$, we first estimate a sequence of depth maps $\mathbf{D} = \{D_i\}_{i=1}^n$ using monocular depth prediction models, with each $D_i \in \mathbb{R}^{\text{H} \times \text{W}}$. Using camera intrinsics $\mathbf{K} \in \mathbb{R}^{3 \times 3}$, we lift each RGB-D pair $(\mathbf{I}_i, D_i)$ into a point cloud $\mathbf{P}_i \in \mathbb{R}^{3 \times (\text{H} \cdot \text{W})}$ via unprojection function $\Pi^{-1}(\cdot)$, forming a dynamic point cloud sequence $\mathbf{P} = \{\mathbf{P}_i\}_{i=1}^n$:

$$\mathbf{P}_i = \Pi^{-1}(\mathbf{I}_i, D_i, \mathbf{K}), \tag{3}$$

where $\Pi^{-1}$ denotes inverse projection from 2D image space to 3D camera space. Next, we define a set of target camera poses $\mathbf{T} = \{\mathbf{T}_i\}_{i=1}^n$, where each $\mathbf{T}_i \in \mathbb{R}^{4 \times 4}$ represents the desired relative transformation from the source view. Using these poses, we render a novel view sequence $\mathbf{I}' = \{\mathbf{I}'_i\}_{i=1}^n$ from the transformed point clouds via forward projection $\Pi(\cdot)$:

$$\mathbf{I}'_i = \Pi(\mathbf{T}_i \cdot \mathbf{P}_i, \mathbf{K}), \tag{4}$$

where $\Pi$ is the standard perspective projection from 3D points to the image plane. In addition to the rendered novel views $\mathbf{I}'$, we generate corresponding visibility masks $\mathbf{M}' = \{\mathbf{M}'_i\}_{i=1}^n$ to capture occluded or out-of-frame regions resulting from the new camera trajectory.

## 4.4 Stochastic Latent Modulation (SLM)

Having addressed the fidelity aspect of dynamic view synthesis, we now turn to the second core requirement: completing regions that become newly visible as the camera moves. As shown in Fig.3, we apply DDIM inversion to videos rendered under novel camera trajectories and interpolate between the VAE-encoded latent $\mathbf{x}_0$ and the DDIM-inverted latent $\epsilon^{\text{inv}}$ using Adaptive K-RNR. However, regions that are occluded in the rendered input remain occluded in both $\mathbf{x}_0$ and $\epsilon^{\text{inv}}$, causing these areas to be regenerated as black in the output.

To investigate this limitation, we examine whether the base model possesses a meaningful physical understanding of the scene that allows it to plausibly infer content in unseen regions. We conduct an analysis on 100 randomly sampled videos from the OpenVid dataset [41], with a particular focus on cases where the input render video lies outside the training distribution or violates basic physical realism. **The central question is whether the model can still produce outputs that are plausible and consistent with the rules of the physical world**. Although unseen areas are also encoded occluded in the inverted latent, we keep $\epsilon^{\text{inv}}$ unchanged, as it retains semantic cues due to attention across visible tokens during the forward trajectory. Instead, we modify the rendered frames by experimenting with different occlusion-filling strategies. As illustrated in Fig.7(c), one approach involves repeating a background patch across the occluded regions. When passed through the 3D VAE and combined with $\epsilon^{\text{inv}}$ through K-RNR, this leads to plausible propagation of visual information into previously unseen areas, as shown in Fig.7(d) with visible visual artifacts.

Motivated by this discovery, we propose **Stochastic Latent Modulation**, where instead of completing unseen regions at the input level, we perform stochastic modulation directly in the latent space. Specifically, given a binary occlusion mask $\mathbf{M} \in \{0,1\}^{\text{B} \times \text{F} \times \text{C} \times \text{H} \times \text{W}}$, where $\mathbf{M} = 1$ indicates occluded regions, and a depth-based background mask $\mathbf{D} \in \{0,1\}^{\text{B} \times \text{F} \times \text{C} \times \text{H} \times \text{W}}$, where $\mathbf{D} = 1$

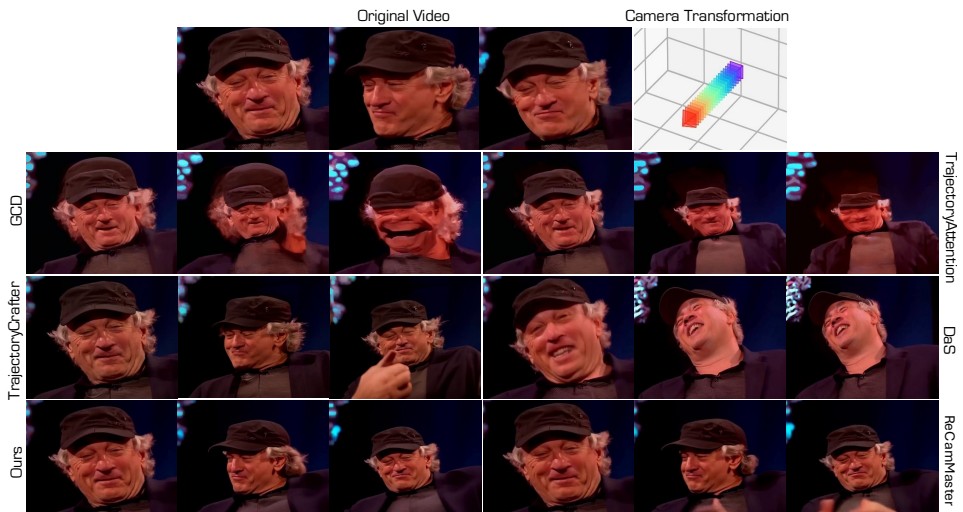

Figure 8: **Qualitative Comparison.** K-RNR with SLM better preserves subject identity and ensures that synthesized regions remain consistent with the original scene.

| Method | Visual Quality | | | | Camera Accuracy | | View Synchronization | | |
|---|---|---|---|---|---|---|---|---|---|
| | FID ↓ | FVD ↓ | CLIP-T ↑ | CLIP-F ↑ | RotErr ↓ | TransErr ↓ | Mat. Pix. (K) ↑ | FVD-V ↓ | CLIP-V ↑ |
| GCD | 89.12 | 482.73 | 28.64 | 91.02 | 3.67 | 6.12 | 603.25 | 429.52 | 82.45 |
| TrajectoryAttention | 78.91 | 342.19 | 30.53 | 93.67 | 3.09 | 5.64 | 620.83 | 310.78 | 84.21 |
| DaS | 71.44 | 201.83 | 32.21 | 96.03 | 2.72 | 5.21 | 638.77 | 182.41 | 86.72 |
| TrajectoryCrafter | 62.77 | 162.67 | 34.13 | 97.48 | 2.39 | 4.89 | 823.91 | 108.38 | 88.36 |
| ReCamMaster | 58.12 | 118.82 | 35.02 | **98.89** | 1.46 | 4.52 | 863.54 | 82.66 | 89.91 |
| Ours | **53.15** | **103.44** | **35.37** | 98.54 | **1.31** | **4.33** | **881.43** | **75.17** | **92.04** |

Table 1: Quantitative comparison of visual quality, camera pose accuracy, and view synchronization on 1000 randomly selected samples from the OpenVid-1M [41] dataset.

marks background areas, we define a visibility-aware sampling mask as $\mathbf{S} = (1 - \mathbf{M}) \cdot \mathbf{D}$ which identifies spatial locations that are both visible and lie on background surfaces. We define a stochastic permutation operator $\mathcal{P}_{\mathbf{S}} : \mathbb{R}^{B \times F \times C \times H \times W} \to \mathbb{R}^{B \times F \times C \times H \times W}$ that samples latent values from positions indicated by $\mathbf{S}$ and randomly redistributes them to the occluded positions indicated by $\mathbf{M}$. Our modulation function is given by $\tilde{\mathbf{x}}_0 = \mathcal{P}_{\mathbf{S}}(\mathbf{x}_0), \quad \tilde{\boldsymbol{\epsilon}}^{\text{inv}} = \mathcal{P}_{\mathbf{S}}(\boldsymbol{\epsilon}^{\text{inv}})$ where $\tilde{\mathbf{x}}$ and $\tilde{\boldsymbol{\epsilon}}$ are the modulated content and noise latents. This operation stochastically fills occluded regions in latent space with contextually relevant signals sampled from visible background areas, enabling the model to synthesize plausible completions aligned with physical scene structure.

## 5 Experiments

**Implementation.** Our framework is built on the pretrained CogVideoX-5B-I2V model. Inference is performed with 50 steps at a strength of 0.95 to ensure $\bar{a}_T > 0$. For all quantitative evaluations, we set the classifier-free guidance (CFG) scale to 6.0 and use a recursion order of $k = 10$ and adaptive order of $\delta = 3$. 3D dynamic point clouds are generated using DepthCrafter [21], following the procedure described in [65]. We apply DDIM inversion with a positive terminal-SNR noise schedule using 30 steps, and adopt v-prediction in all cases. For quantitative evaluations, we use CogVideoX's modified DDIM sampling method in the reverse trajectory. The output resolution is fixed at $480 \times 720$, and all experiments are conducted on a single NVIDIA L40 GPU.

**Evaluation Set.** We construct a dataset of 1100 videos to evaluate performance across varying content and motion complexity: 1000 from OpenVid-1M [41], 50 from DAVIS [43], and 50 AI-generated videos. OpenVid-1M provides semantically rich scenes, DAVIS offers high-motion content for testing temporal stability, and AI-generated samples assess generalization to synthetic inputs. Each video is rendered under 10 canonical camera trajectories including transla-

| Method | FID ↓ | CLIP-T ↑ | CLIP-V ↑ | PSNR ↑ |
|---|---|---|---|---|
| Random Noise | 74.86 | **37.12** | 73.74 | 12.06 |
| DDIM Inversion | 102.54 | 19.98 | 63.39 | 5.43 |
| + K-RNR w.o AS | 71.80 | 31.25 | 86.78 | 14.99 |
| + K-RNR w AS | 61.43 | 33.46 | 89.12 | 15.64 |
| + K-RNR w SLM | **53.15** | 35.37 | **92.04** | **16.28** |

Figure 9: Ablation on K-RNR, Adaptive Scaling, and Stochastic Latent Modulation

| Method | PSNR↑ | | | | SSIM↑ | | | | LPIPS↓ | | | |
|---|---|---|---|---|---|---|---|---|---|---|---|---|
| | OpenVid [41] | DAVIS [43] | Synthetic | **Mean** | OpenVid [41] | DAVIS [43] | Synthetic | **Mean** | OpenVid [41] | DAVIS [43] | Synthetic | **Mean** |
| GCD | 9.87 | 8.32 | 10.57 | 9.58 | 0.212 | 0.191 | 0.227 | 0.210 | 0.739 | 0.754 | 0.681 | 0.724 |
| TrajectoryAttention | 10.11 | 9.70 | 11.04 | 10.28 | 0.241 | 0.211 | 0.272 | 0.241 | 0.685 | 0.708 | 0.618 | 0.670 |
| DaS | 11.37 | 10.14 | 12.27 | 11.26 | 0.309 | 0.259 | 0.348 | 0.305 | 0.586 | 0.621 | 0.545 | 0.584 |
| TrajectoryCrafter | 13.02 | 10.89 | 13.94 | 12.61 | 0.428 | 0.306 | 0.501 | 0.411 | 0.366 | 0.646 | 0.537 | 0.516 |
| ReCamMaster | 15.84 | 11.31 | 14.17 | 13.77 | 0.610 | 0.339 | **0.623** | 0.524 | 0.421 | 0.588 | 0.517 | 0.508 |
| Ours | **16.28** | **12.64** | **14.59** | **14.50** | **0.623** | **0.354** | 0.617 | **0.531** | **0.397** | **0.561** | **0.504** | **0.487** |

Table 2: Quantitative comparison on our curated benchmark. We report PSNR (↑), SSIM (↑), and LPIPS (↓), averaged over 10 canonical camera trajectories per video.

tions, pans, tilts, and arcs, to evaluate robustness under diverse viewpoint shifts.

**Comparison Baselines.** We compare our method against five baselines: GCD [52], TrajectoryAttention [59], RecamMaster [2], TrajectoryCrafter [65], and Diffusion-as-Shader (DaS) [17]. GCD and TrajectoryAttention are built on SVD [4], RecamMaster is based on Wan [53], while TrajectoryCrafter, DaS, and our method are based on CogVideoX.

**Evaluation Metrics.** We evaluate our method for camera pose accuracy, source-target synchronization, and visual quality. For camera accuracy, we use GLOMAP [42] to extract estimated camera trajectories and report rotation and translation errors (RotErr, TransErr) following [18, 2]. Synchronization is measured using GIM [48] by counting matched pixels with high confidence (Mat. Pix.), along with FVD-V [60] and CLIP-V [28], which compute CLIP similarity between source and target frames at corresponding timestamps. Visual quality is evaluated using FID [19], FVD [51], CLIP-T, and CLIP-F, capturing fidelity, text alignment, and temporal consistency, respectively. We additionally compute the full reference metrics PSNR, SSIM, and LPIPS on the OpenVid-1M, DAVIS, and Sora-generated videos [5] to quantify per-frame visual fidelity with respect to ground truth frames.

**Main Results.** As reported in Table 1, our method achieves state-of-the-art performance across all quantitative evaluation axes, encompassing visual fidelity, camera pose accuracy, and view synchronization. The results demonstrate that our framework consistently preserves semantic content and visual coherence while maintaining accurate geometric alignment under camera transformations. Compared to existing baselines, our approach yields improved consistency across frames and more precise reconstruction of dynamic scenes, validating the effectiveness of our noise-space formulation. Furthermore, Table 2 reports full-reference metrics, where our method exhibits robust reconstruction quality across diverse datasets and camera trajectories, further confirming its generalizability and resilience under varying content complexity and motion dynamics. We show identity preservation quality of our method and the baselines in Fig.8. Our framework produces visually coherent results under various viewpoints and demonstrates strong temporal alignment with the source footage. For video samples, please refer to the Supplementary Material.

**Ablation Studies.** Table 9 shows the impact of our proposed methods: K-RNR, Adaptive K-RNR, and K-RNR with Stochastic Latent Modulation. Directly using DDIM-inverted latents leads to poor results, often producing oversaturated and washed-out outputs, as seen in Fig. 2(d). Initializing with random noise also results in weak view synchronization. In contrast, our methods significantly improve both view alignment and reconstruction quality, as reflected in PSNR and FID scores.

**Noise Initialization Ablations.** The results presented in Figure 10 provide a comparative evaluation of various initialization strategies for video reconstruction in the absence of camera transformations. The baseline method that begins generation with standard normal noise ($\epsilon$) underperforms across all metrics, which is expected due to the lack of structured guidance during synthesis. Injecting signal

| Method | PSNR↑ | SSIM↑ | LPIPS↓ |
|---|---|---|---|
| Random Noise | 12.03 | 0.313 | 0.486 |
| Encoded Video + Random Noise | 15.97 | 0.674 | 0.539 |
| DDIM Inversion | 9.08 | 0.315 | 0.904 |
| Encoded Video + DDIM Inversion | 10.16 | 0.324 | 0.907 |
| Random Noise + KV Caching | 23.98 | 0.824 | 0.118 |
| **K-RNR** | **29.56** | **0.910** | **0.063** |

Figure 10: Ablation on noise initialization strategies for video reconstruction without camera transformation.

via a linear combination of VAE-encoded video latents ($\mathbf{x}_0$) and noise, as in the Encoded Video + Random Noise strategy, yields noticeable improvements, indicating the benefit of directly in-

corporating source video content into the initial conditions. In contrast, DDIM Inversion, which initializes with an inverted latent but without scheduler-consistent interpolation, achieves the lowest reconstruction quality, yielding high saturation, washed-out generations. The marginal improvement obtained by combining the encoded latent with DDIM inversion further underscores the sensitivity of the diffusion trajectory to initialization fidelity.

Random Noise + KV Caching introduces a mechanism where the generation initiated from noise is guided by attending to key-value pairs derived from a parallel DDIM-inverted path, integrating cross-stream structural memory. This strategy shows some gains, particularly in perceptual quality as measured by LPIPS with the expense of reduced efficiency since 2 parallel attention computation over the extended sequence dimension is performed. Our proposed K-RNR approach that achieves the highest performance across all metrics, with PSNR, SSIM, and LPIPS values of 29.56, 0.910, and 0.063 respectively. These results confirm the effectiveness of recursive noise representation for high-fidelity video reconstruction. The superior quantitative outcomes suggest that K-RNR is capable of leveraging structured priors in noise space more effectively than existing baselines. Results are demonstrated in Figure 2 and corresponding videos are shared in `website.html`

**Discrete K-order Ablations.** Figure 11 presents an ablation study on the discrete recursion depth $k$ in K-RNR, following the application of adaptive scaling. The results demonstrate a clear performance trend as $k$ increases. For shallow recursion depths ($k = 1$ and $k = 2$), the model exhibits poor reconstruction quality across all metrics, indicating that insufficient recursive refinement fails to recover meaningful structure in the video content. A substantial performance jump is observed at $k = 3$, suggesting that a minimum level of recursive processing is necessary to capture the underlying temporal and spatial consistency required for high-fidelity generation.

| $K$-Depth | PSNR $\uparrow$ | SSIM $\uparrow$ | LPIPS $\downarrow$ |
|---|---|---|---|
| $k = 1$ | 7.82 | 0.221 | 0.896 |
| $k = 2$ | 8.85 | 0.231 | 0.871 |
| $k = 3$ | 15.91 | 0.550 | **0.465** |
| $k = 4$ | 15.94 | 0.550 | 0.468 |
| $k = 5$ | 16.00 | 0.550 | 0.489 |
| $k = 6$ | **16.39** | 0.555 | 0.471 |
| $k = 7$ | 16.34 | **0.558** | 0.474 |
| $k = 8$ | 15.30 | 0.545 | 0.483 |

Figure 11: Ablation on the recursion depth $k$ in K-RNR after applying adaptive scaling.

As $k$ increases beyond 3, PSNR and SSIM metrics improve steadily, peaking at $k = 6$ and $k = 7$ respectively. The LPIPS metric reaches its lowest value at $k = 3$ (0.465), indicating optimal perceptual similarity at moderate recursion depth, though values remain competitive through $k = 7$. Notably, performance begins to degrade at $k = 8$, likely due to over-recursion, which may introduce noise or overfitting artifacts into the refinement process. These findings suggest that while increasing recursion depth generally enhances reconstruction, there exists a sweet spot around $k = 6$ to $k = 7$ that balances iterative refinement with stability. This trade-off is essential to consider when tuning K-RNR for optimal video reconstruction performance.

## 6 Discussion

**Limitations and Broader Impact** Our method provides a training-free framework for generative camera control in real-world videos, making it broadly accessible for creative editing. However, it inherits biases from the base diffusion model which may limit performance in scenes with uncommon objects, or heavy occlusion. Stochastic latent modulation can also produce unstable or incoherent results when large regions become newly visible. The ability to generate realistic synthetic content raises concerns, highlighting the need for future safeguards such as attribution or model auditing.

**Conclusion** In this paper, we introduce a training-free framework for dynamic view synthesis from monocular videos. Our key contributions (1) the identification of the Zero-Terminal SNR Collapse Problem, (2) the development of the K-order Recursive Noise Representation for the use of deterministic inversion, and (3) the Stochastic Latent Modulation technique for occlusion-aware scene completion. Together, they enable high-fidelity synthesis of novel views without fine-tuning or architectural changes. Through rigorous theoretical analysis and empirical validation, we demonstrate that structured manipulation of the noise space alone can unlock new capabilities in generative models, offering a principled and practical path toward controllable, efficient dynamic scene generation.

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

# Table of Contents

## A   Videos and Website

To facilitate comprehensive evaluation and enhance result accessibility, we provide 100+ video results including motivation examples, qualitative results, ablation studies, qualitative comparisons, and limitations in our project page.

## B   Symbols and Notations

In this section, we present the symbols and notations used throughout the paper to ensure clarity and consistency in our mathematical and algorithmic descriptions.

## C   Elaboration on Proposition 4.1

Consider a variance-preserving noise schedule $\{\alpha_t\}_{t=0}^{T}$ with cumulative products defined as $\bar{\alpha}_t = \prod_{s=1}^{t}(1 - \beta_s)$, where the schedule enforces a zero terminal signal-to-noise ratio (SNR), such that

| Symbol | Description |
|--------|-------------|
| **Video and Frame Symbols** | |
| $\mathbf{V}$ | Source video |
| $\mathbf{I}_i$ | Individual frame of the source video |
| $\mathbf{D} = \{D_i\}_{i=1}^n$ | Sequence of depth maps |
| $D_i$ | Depth map for frame $\mathbf{I}_i$ |
| $\mathbf{K}$ | Camera intrinsics matrix |
| $\mathbf{P}_i$ | Point cloud for frame $\mathbf{I}_i$ |
| $\mathbf{T}_i$ | Target camera pose for frame $i$ |
| $\mathbf{I}'_i$ | Rendered novel view for frame $i$ |
| $\mathbf{M}'$ | Visibility masks for novel views |
| **Latent Space Symbols** | |
| $\Phi_t$ | Diffusion for timestep $t$ |
| $\alpha_t$ | Cumulative signal coefficient at timestep $t$ |
| $\bar{\alpha}_t$ | Cumulative product of $\alpha_t$ |
| $\mathbf{x}_{\text{init}}$ | Initial noise for diffusion process |
| $\mathbf{x}_0$ | VAE-encoded latent also our pivot latent |
| $\epsilon$ | Noise sample from standard normal distribution |
| $\epsilon^{\text{inv}}$ | DDIM inverted latent |
| $\epsilon^{(k)}$ | K-order recursive noise representation |
| **Mask and Modulation Symbols** | |
| $\mathbf{M}$ | Binary occlusion mask |
| $\mathbf{D}$ | Depth-based near depth mask |
| $\mathbf{S}$ | Visibility-aware sampling mask |
| $\mathcal{P}_\mathbf{S}$ | Stochastic permutation operator |
| $\tilde{\mathbf{x}}_0$ | Modulated content latent |
| $\hat{\epsilon}^{\text{inv}}$ | Modulated noise latent |

Table 1: List of symbols used in the paper.

$\bar{\alpha}_T = 0$. The forward diffusion map is given by:

$$\Phi_T(x_0, \epsilon) = \sqrt{\bar{\alpha}_T}x_0 + \sqrt{1 - \bar{\alpha}_T}\epsilon,$$

where $x_0 \in \mathbb{R}^{\text{F}\times\text{C}\times\text{H}\times\text{W}}$ is the initial latent variable, and $\epsilon \sim \mathcal{N}(0, I)$ is a noise sample drawn from a standard normal distribution.

## C.1  Forward Diffusion Map Under Zero-Terminal SNR

Since the zero-terminal SNR noise schedule specifies $\bar{\alpha}_T = 0$, substitute this into the definition of $\Phi_T$:

$$\Phi_T(x_0, \epsilon) = \sqrt{\bar{\alpha}_T}x_0 + \sqrt{1 - \bar{\alpha}_T}\epsilon = \sqrt{0}x_0 + \sqrt{1 - 0}\epsilon = 0 \cdot x_0 + 1 \cdot \epsilon = \epsilon.$$

Thus, $\Phi_T(x_0, \epsilon) = \epsilon$, which depends solely on the noise $\epsilon$ and is independent of the initial latent $x_0$. For any two initial latents $x_0, x'_0 \in \mathbb{R}^{\text{F}\times\text{C}\times\text{H}\times\text{W}}$ and a fixed noise sample $\epsilon$, it follows that:

$$\Phi_T(x_0, \epsilon) = \epsilon \quad \text{and} \quad \Phi_T(x'_0, \epsilon) = \epsilon.$$

Therefore, $\Phi_T(x_0, \epsilon) = \Phi_T(x'_0, \epsilon) = \epsilon$, regardless of whether $x_0 = x'_0$ or $x_0 \neq x'_0$.

## C.2  Breakdown of Injectivity

A function $f : A \rightarrow B$ is injective if, for all $a, a' \in A$, $f(a) = f(a')$ implies $a = a'$. Consider the map $\Phi_T(\cdot, \epsilon) : \mathbb{R}^{\text{F}\times\text{C}\times\text{H}\times\text{W}} \rightarrow \mathbb{R}^{\text{F}\times\text{C}\times\text{H}\times\text{W}}$ with $\epsilon$ fixed. From §C.1, for any distinct $x_0, x'_0 \in \mathbb{R}^{\text{F}\times\text{C}\times\text{H}\times\text{W}}$ where $x_0 \neq x'_0$, we have:

$$\Phi_T(x_0, \epsilon) = \epsilon = \Phi_T(x'_0, \epsilon).$$

Since $\Phi_T(x_0, \epsilon) = \Phi_T(x'_0, \epsilon)$ holds even when $x_0 \neq x'_0$, the condition for injectivity is violated. Hence, $\Phi_T(\cdot, \epsilon)$ is not injective in $x_0$, as multiple (indeed, all) initial latents $x_0$ map to the same output $\epsilon$ for a given $\epsilon$.

## C.3 Implications for Deterministic Inversion

In diffusion models, the terminal state is denoted $x_T = \Phi_T(x_0, \epsilon)$, which, under the condition $\bar{\alpha}_T = 0$, simplifies to $x_T = \epsilon$. Deterministic inversion methods, such as DDIM inversion, aim to recover the original latent $x_0$ from $x_T$ by reversing the forward diffusion process. These methods assume that the forward map $\Phi_T$ can be inverted uniquely, which requires $\Phi_T$ to be injective. However, since $\Phi_T(\cdot, \epsilon)$ is not injective, multiple distinct $x_0$ produce the same $x_T = \epsilon$. Consequently, given only $x_T$, it is impossible to determine which $x_0$ among the infinitely many possible initial latents was the original, rendering unique recovery via deterministic inversion unfeasible.

## D Elaboration on Proposition 4.2

In this section, we prove the closed-form expressions associated with the recursive noise initialization process K-RNR outlined in Proposition 4.2. The recursive process is defined as follows: for an initial step where $k = 1$, the expression is given by

$$\epsilon^{(1)} = \sqrt{\bar{\alpha}_t}x_0 + \sqrt{1 - \bar{\alpha}_t}\epsilon^{\mathrm{inv}},$$

and for subsequent steps where $k > 1$, the expression becomes

$$\epsilon^{(k)} = \sqrt{\bar{\alpha}_t}x_0 + \sqrt{1 - \bar{\alpha}_t}\epsilon^{(k-1)}.$$

Here, $x_0 \in \mathbb{R}^{\mathrm{F \times C \times H \times W}}$ represents the pivot latent variable, $\bar{\alpha}_t > 0$ denotes the cumulative signal coefficient at timestep $t$, and $\epsilon^{\mathrm{inv}}$ is the initial noise term.

The proposition posits two closed-form expressions. For the discrete recursion depth, where $k \in \mathbb{N}_{\geq 0}$, the expression is

$$\epsilon^{(k)} = \left( \sum_{i=1}^{k} \sqrt{\bar{\alpha}_t} \left( \sqrt{1 - \bar{\alpha}_t} \right)^{i-1} \right) x_0 + \left( \sqrt{1 - \bar{\alpha}_t} \right)^{k} \epsilon^{\mathrm{inv}}.$$

For the continuous recursion depth, where $k \in \mathbb{R}_{\geq 0}$, the expression is

$$\epsilon^{(k)} = \left( \sqrt{\bar{\alpha}_t} \frac{1 - \left( \sqrt{1 - \bar{\alpha}_t} \right)^{k}}{1 - \sqrt{1 - \bar{\alpha}_t}} \right) x_0 + \left( \sqrt{1 - \bar{\alpha}_t} \right)^{k} \epsilon^{\mathrm{inv}}.$$

The proof is divided into two parts: the discrete case is addressed in §D.1, and continuous case is addressed in §D.2.

### D.1 Proof for the Discrete Case: $k \in \mathbb{N}_{\geq 0}$

To verify the closed-form expression for discrete values of $k$, mathematical induction is employed as a method of proof.

For the initial step, consider the case where $k = 1$. The recursive definition states that

$$\epsilon^{(1)} = \sqrt{\bar{\alpha}_t}x_0 + \sqrt{1 - \bar{\alpha}_t}\epsilon^{\mathrm{inv}}.$$

To confirm this, the proposed closed-form expression is evaluated at $k = 1$:

$$\epsilon^{(1)} = \left( \sum_{i=1}^{1} \sqrt{\bar{\alpha}_t} \left( \sqrt{1 - \bar{\alpha}_t} \right)^{i-1} \right) x_0 + \left( \sqrt{1 - \bar{\alpha}_t} \right)^{1} \epsilon^{\mathrm{inv}}.$$

The summation involves only one term, corresponding to $i = 1$. This term is calculated as follows:

$$\sqrt{\bar{\alpha}_t} \left( \sqrt{1 - \bar{\alpha}_t} \right)^{1-1} = \sqrt{\bar{\alpha}_t} \left( \sqrt{1 - \bar{\alpha}_t} \right)^{0} = \sqrt{\bar{\alpha}_t} \cdot 1 = \sqrt{\bar{\alpha}_t}.$$

Thus, the closed-form expression becomes

$$\epsilon^{(1)} = \sqrt{\bar{\alpha}_t}x_0 + \sqrt{1 - \bar{\alpha}_t}\epsilon^{\mathrm{inv}},$$

which is identical to the recursive definition. This establishes the validity of the expression for the base case.

Next, suppose that for some positive integer $n \geq 1$, the closed-form expression holds true:

$$\epsilon^{(n)} = \left( \sum_{i=1}^{n} \sqrt{\bar{\alpha}_t} \left( \sqrt{1 - \bar{\alpha}_t} \right)^{i-1} \right) x_0 + \left( \sqrt{1 - \bar{\alpha}_t} \right)^{n} \epsilon^{\text{inv}}.$$

The objective is now to demonstrate that this expression remains valid for the next integer, $k = n + 1$. According to the recursive definition,

$$\epsilon^{(n+1)} = \sqrt{\bar{\alpha}_t} x_0 + \sqrt{1 - \bar{\alpha}_t} \epsilon^{(n)}.$$

The inductive hypothesis is substituted into this equation, yielding

$$\epsilon^{(n+1)} = \sqrt{\bar{\alpha}_t} x_0 + \sqrt{1 - \bar{\alpha}_t} \left[ \left( \sum_{i=1}^{n} \sqrt{\bar{\alpha}_t} \left( \sqrt{1 - \bar{\alpha}_t} \right)^{i-1} \right) x_0 + \left( \sqrt{1 - \bar{\alpha}_t} \right)^{n} \epsilon^{\text{inv}} \right].$$

The factor $\sqrt{1 - \bar{\alpha}_t}$ is applied to each term within the brackets. For the summation term, this results in

$$\sqrt{1 - \bar{\alpha}_t} \cdot \sum_{i=1}^{n} \sqrt{\bar{\alpha}_t} \left( \sqrt{1 - \bar{\alpha}_t} \right)^{i-1} = \sum_{i=1}^{n} \sqrt{\bar{\alpha}_t} \left( \sqrt{1 - \bar{\alpha}_t} \right)^{i},$$

and for the noise term,

$$\sqrt{1 - \bar{\alpha}_t} \cdot \left( \sqrt{1 - \bar{\alpha}_t} \right)^{n} = \left( \sqrt{1 - \bar{\alpha}_t} \right)^{n+1}.$$

Thus, the expression for $\epsilon^{(n+1)}$ is written as

$$\epsilon^{(n+1)} = \sqrt{\bar{\alpha}_t} x_0 + \left( \sum_{i=1}^{n} \sqrt{\bar{\alpha}_t} \left( \sqrt{1 - \bar{\alpha}_t} \right)^{i} \right) x_0 + \left( \sqrt{1 - \bar{\alpha}_t} \right)^{n+1} \epsilon^{\text{inv}}.$$

The terms involving $x_0$ are then grouped together:

$$\epsilon^{(n+1)} = \left( \sqrt{\bar{\alpha}_t} + \sum_{i=1}^{n} \sqrt{\bar{\alpha}_t} \left( \sqrt{1 - \bar{\alpha}_t} \right)^{i} \right) x_0 + \left( \sqrt{1 - \bar{\alpha}_t} \right)^{n+1} \epsilon^{\text{inv}}.$$

To express this as a single summation, it is noted that $\sqrt{\bar{\alpha}_t}$ can be written as $\sqrt{\bar{\alpha}_t} \left( \sqrt{1 - \bar{\alpha}_t} \right)^{0}$. This allows the expression to be rewritten by adjusting the summation indices:

$$\sqrt{\bar{\alpha}_t} + \sum_{i=1}^{n} \sqrt{\bar{\alpha}_t} \left( \sqrt{1 - \bar{\alpha}_t} \right)^{i} = \sum_{i=0}^{n} \sqrt{\bar{\alpha}_t} \left( \sqrt{1 - \bar{\alpha}_t} \right)^{i}.$$

This summation from $i = 0$ to $n$ corresponds exactly to the desired form when re-indexed:

$$\sum_{i=0}^{n} \sqrt{\bar{\alpha}_t} \left( \sqrt{1 - \bar{\alpha}_t} \right)^{i} = \sum_{i=1}^{n+1} \sqrt{\bar{\alpha}_t} \left( \sqrt{1 - \bar{\alpha}_t} \right)^{i-1},$$

since each term aligns appropriately with the change in index. Therefore, the expression becomes

$$\epsilon^{(n+1)} = \left( \sum_{i=1}^{n+1} \sqrt{\bar{\alpha}_t} \left( \sqrt{1 - \bar{\alpha}_t} \right)^{i-1} \right) x_0 + \left( \sqrt{1 - \bar{\alpha}_t} \right)^{n+1} \epsilon^{\text{inv}},$$

which matches the proposed closed-form expression for $k = n + 1$. This step confirms the inductive hypothesis for the next integer, and by the principle of mathematical induction, the closed-form expression is valid for all positive integers $k$ which completes the proof ∎

### D.2   Proof for the Continuous Case: $k \in \mathbb{R}_{\geq 0}$

To extend the result to real values of $k$, the discrete case's summation is analyzed as a geometric series. Let the ratio $r = \sqrt{1 - \bar{\alpha}_t}$, where, given $0 < \bar{\alpha}_t < 1$, it follows that $0 < r < 1$. The summation in the discrete expression is expressed as

$$\sum_{i=1}^{k} \sqrt{\bar{\alpha}_t} r^{i-1} = \sqrt{\bar{\alpha}_t} \sum_{i=0}^{k-1} r^{i}.$$

The formula for the sum of a finite geometric series is applied here:

$$\sum_{i=0}^{k-1} r^i = \frac{1 - r^k}{1 - r}.$$

This allows the summation to be rewritten as

$$\sqrt{\bar{\alpha}_t} \sum_{i=0}^{k-1} r^i = \sqrt{\bar{\alpha}_t} \cdot \frac{1 - r^k}{1 - r}.$$

Substituting $r = \sqrt{1 - \bar{\alpha}_t}$ back into the expression, it becomes

$$\sqrt{\bar{\alpha}_t} \cdot \frac{1 - \left(\sqrt{1 - \bar{\alpha}_t}\right)^k}{1 - \sqrt{1 - \bar{\alpha}_t}}.$$

Incorporating this into the discrete closed-form expression, the result is

$$\epsilon^{(k)} = \left( \sqrt{\bar{\alpha}_t} \cdot \frac{1 - \left(\sqrt{1 - \bar{\alpha}_t}\right)^k}{1 - \sqrt{1 - \bar{\alpha}_t}} \right) x_0 + \left(\sqrt{1 - \bar{\alpha}_t}\right)^k \epsilon^{\text{inv}}.$$

This formulation is well-defined for all real $k \geq 0$, as the exponential terms are continuous functions over the real numbers which completes the proof ■

# E   Elaboration on Stochastic Latent Modulation

In this section, we provide a detailed technical elaboration of the Stochastic Latent Modulation (SLM) mechanism, a key component of our approach to dynamic view synthesis. SLM addresses the challenge of synthesizing plausible content for regions that become newly visible due to camera motion, operating directly in the latent space of a pre-trained video diffusion model. This process modulates both the VAE-encoded latent $\mathbf{x}_0$ and the inverted latent $\epsilon^{\text{inv}}$ using a single binary occlusion mask and depth map, ensuring a consistent and efficient strategy for handling occlusions. By leveraging visibility-aware sampling and stochastic permutation, SLM enables the diffusion model to infer content for occluded regions without requiring architectural changes or additional training.

## E.1   Technical Details of Stochastic Latent Modulation

The SLM process modulates the latents $\mathbf{x}$ and $\epsilon$ by filling their occluded regions with values sampled from visible, depth-specific areas, using a single mask $\mathbf{M}$ and depth map $\mathbf{D}$ to guide the operation. This begins with the computation of a visibility mask, defined as $\mathbf{V} = (1 - \mathbf{M}) \cdot (\mathbf{D})$, which identifies regions that are both visible (where $\mathbf{M} = 0$) and depthwise near (where $\mathbf{D}$). These regions serve as the source pool for sampling, as they contain stable and contextually relevant latent values from the scene. The target regions, where content synthesis is needed, correspond to the occluded areas where $\mathbf{M} = 1$.

The modulation proceeds by identifying the spatial indices of the source and target regions. The set of source indices, $\mathcal{I}_{\text{source}}$, consists of all positions where $\mathbf{V} = 1$, while the set of target indices, $\mathcal{I}_{\text{target}}$, includes all positions where $\mathbf{M} = 1$. For each latent, SLM counts the number of occluded elements (i.e., the size of $\mathcal{I}_{\text{target}}$) and randomly selects an equal number of indices from $\mathcal{I}_{\text{source}}$. These randomly chosen source values are then assigned to the target positions. Specifically, for $\mathbf{x}$, the values at indices $\mathbf{i} \in \mathcal{I}_{\text{target}}$ are replaced with values from randomly selected indices $\mathbf{j} \in \mathcal{I}_{\text{source}}$, such that $\mathbf{x_i} = \mathbf{x_j}$. The same process is applied to $\epsilon$, where $\epsilon_{\mathbf{i}} = \epsilon_{\mathbf{j}}$ for corresponding pairs of indices. This stochastic sampling ensures that the occluded regions of both latents are populated with plausible content drawn from the visible, near-depth areas of the scene.

The use of a single mask and depth map for both $\mathbf{x}$ and $\epsilon$ ensures that the source and target regions remain consistent across the two latents, while the independent application of the sampling process to each latent preserves their distinct roles in the diffusion pipeline. The randomness in selecting source indices introduces variability, allowing the diffusion model to explore diverse completions for the occluded regions, all while maintaining coherence with the visible parts of the scene.

### E.2 Algorithm for Stochastic Latent Modulation

---

**Algorithm 1** Stochastic Latent Modulation

---

1: **Input:** $\mathbf{x} \in \mathbb{R}^{B \times F \times C \times H \times W}$, $\boldsymbol{\epsilon} \in \mathbb{R}^{B \times F \times C \times H \times W}$, $\mathbf{M} \in \{0,1\}^{B \times F \times C \times H \times W}$, $\mathbf{D} \in \mathbb{R}^{B \times F \times C \times H \times W}$
2: **Output:** Modulated $\mathbf{x}$, Modulated $\boldsymbol{\epsilon}$
3: Compute visibility mask $\mathbf{V} = (1 - \mathbf{M}) \cdot \mathbf{D}$
4: Let $\mathcal{I}_{\text{source}} = \{\mathbf{i} \mid \mathbf{V_i} = 1\}$
5: Let $\mathcal{I}_{\text{target}} = \{\mathbf{i} \mid \mathbf{M_i} = 1\}$
6: **for** each $\mathbf{i} \in \mathcal{I}_{\text{target}}$ **do**
7:     Sample $\mathbf{j} \sim \text{Uniform}(\mathcal{I}_{\text{source}})$
8:     Set $\boldsymbol{\epsilon_i} = \boldsymbol{\epsilon_j}$
9:     Set $\mathbf{x_i} = \mathbf{x_j}$
10: **end for**
11: **return** $\mathbf{x}$, $\boldsymbol{\epsilon}$

---

## F  More Ablation Studies

In this section, we present additional ablation studies to further analyze the components of our approach. in §F.1, we analyze the role of the adaptive normalization latent depth $\delta$.

### F.1  Adaptive Reference Latent Index $\delta$ Ablations

Figure 1 presents an ablation study on the choice of the adaptive latent index $\delta$, which determines the reference noise level used for adaptive normalization between the $k$-th order noise and the $\delta$-order noise. In all our experiments, we set $\delta = 3$, and the results in this ablation empirically validate this design choice. When $\delta = 3$, the model achieves the highest reconstruction quality across all evaluation metrics, with a PSNR of 24.97, SSIM of 0.885, and LPIPS of 0.078.

| $\delta$ Index | PSNR $\uparrow$ | SSIM $\uparrow$ | LPIPS $\downarrow$ |
|---|---|---|---|
| $\delta = 1$ | 10.32 | 0.342 | 0.883 |
| $\delta = 2$ | 19.23 | 0.748 | 0.148 |
| $\delta = 3$ | **24.97** | **0.885** | **0.078** |
| $\delta = 4$ | 15.29 | 0.592 | 0.240 |
| $\delta = 5$ | 13.92 | 0.468 | 0.329 |
| $\delta = 6$ | 12.66 | 0.333 | 0.451 |
| $\delta = 7$ | 11.28 | 0.244 | 0.604 |

Figure 1: Ablation on the adaptive index $\delta$.

Performance degrades notably when $\delta$ deviates from this setting. For instance, lower values of $\delta$ such as 1 and 2 lead to insufficient regularization, producing reconstructions with low fidelity and poor perceptual quality. Conversely, higher values of $\delta$ (i.e., $\delta \geq 4$) introduce excessive deviation in the normalization reference, which appears to destabilize the refinement process and result in less consistent outputs. This pattern suggests that $\delta = 3$ offers an optimal trade-off by aligning the reference noise distribution closely with the target generation stage, enabling more effective adaptive normalization. These findings confirm that careful selection of the latent reference index is critical for preserving quality in recursive refinement.

## G  Discussion on Quantitative Results

Table 1 and Table 2 in the main paper present a comprehensive quantitative evaluation of our framework against recent methods across multiple axes, including visual quality, camera pose accuracy, view synchronization, and reconstruction fidelity. The baseline methods span three architectural families: GCD and TrajectoryAttention are built upon the Stable Video Diffusion backbone, Diffusion as Shader (DaS) and TrajectoryCrafter share the CogVideoX foundation with our method, and ReCamMaster is based on the Wan architecture.

In our experiments, we observe that methods relying on Stable Video Diffusion, such as GCD and TrajectoryAttention, consistently underperform in preserving the identity and motion dynamics of the original videos when camera transformations are introduced. This can be attributed to the limited expressiveness of the Stable Video Diffusion architecture compared to the more semantically rich representations offered by CogVideoX and Wan. Among the CogVideoX-based approaches, Diffusion

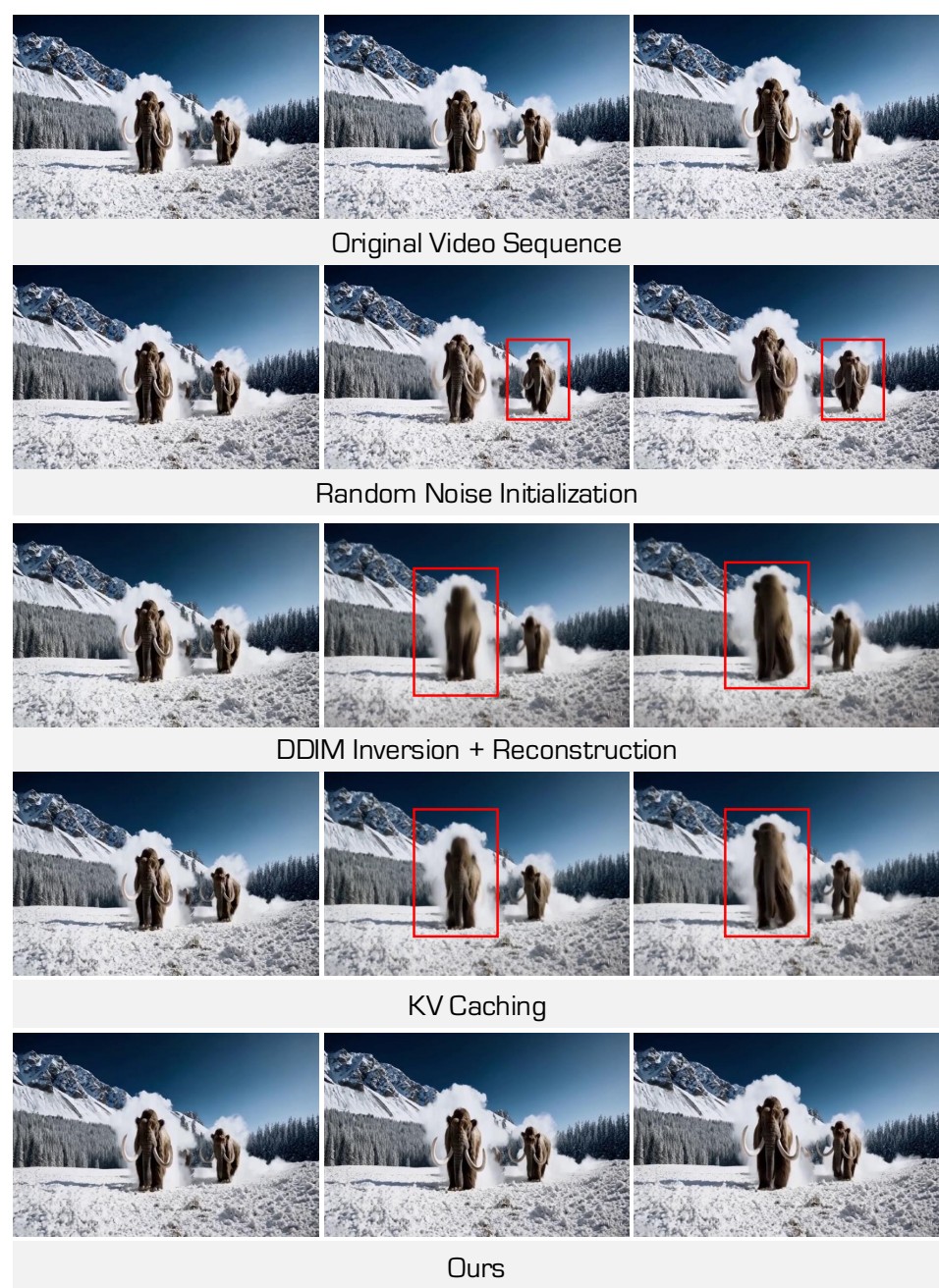

Figure 2: **Video Reconstruction Strategies.** We perform quantitative and qualitative evaluation on video reconstruction without camera transformation application. Video results can be found in the supplementary material.

as Shader struggles to maintain action fidelity, often generating semantically coherent frames that fail to reflect the intended motion trajectory. TrajectoryCrafter achieves a stronger balance between action fidelity and identity preservation; however, we note that identity consistency tends to degrade toward the latter segments of the video. ReCamMaster, while effective in its synthesis, incurs significant inefficiency due to its reliance on concatenating source and target video frames along the frame channel. This design increases the overall token sequence length, which not only limits scalability but also results in considerably slower inference speeds. In contrast, our proposed method retains both high fidelity and identity consistency across the video while maintaining efficient inference. The quantitative comparisons are shared in `website.html`.

# H   Technical Elaborations | Why not set $\bar{\alpha}_T = 0$?

**Q: In experiments we use a strength of 0.95 to ensure $\bar{\alpha}_T > 0$, why not set $\bar{\alpha}_T = 0$?**

When $\bar{\alpha}_T = 0$, Equation (1) reduces to standard DDIM inversion, which is the main motivation of this paper: to demonstrate that standard DDIM inversion does not work under a zero terminal SNR setting.

Let's see this situation step by step. In Equation (1), when $t = T$ where $\bar{\alpha}_T = 0$:

$$\left( \sum_{i=1}^{k} \sqrt{\bar{\alpha}_T} (\sqrt{1 - \bar{\alpha}_T})^{i-1} \right) x_0 + (\sqrt{1 - \bar{\alpha}_T})^k \varepsilon_{\text{inv}}$$

$$= \left( \sum_{i=1}^{k} 0 \times \sqrt{1} \right) x_0 + (\sqrt{1})^k \varepsilon_{\text{inv}}$$

$$= \varepsilon_{\text{inv}}$$

Thus, when $\bar{\alpha}_T = 0$, the entire term collapses to the pure noise term $\varepsilon_{\text{inv}}$, showing that no image content can be reconstructed, precisely why $\bar{\alpha}_T$ should remain nonzero.

# I   Parameter Settings

## I.1   How To Choose $k$?

We obtained the best results when we set $k = 3$ and $k = 6$. Note that we do not tweak the $k$ value per video–camera pair. We also want to clarify an important point:

- **Book reading example:** We presented video results for $k = 20$. This choice was not made because $k = 20$ is optimal, but rather because it represents a relatively high value of $k$. Our goal in that experiment is to highlight the effectiveness of our *adaptive normalization* extension of K-RNR when $k$ is high, which is why we chose to demonstrate the experiment at a higher setting.
- **Monkey example:** We wanted to demonstrate the K-RNR's effect on rendered videos with increasing $k$ values. The logic behind that experiment is demonstrating to readers the *evolution of videos* with different $k$ settings. As stated earlier, $k = 6$ generates plausible results.
- **Elephant and duck examples:** We aimed to demonstrate the effectiveness of K-RNR in source video reconstruction when there is no occlusion (i.e., no SLM involved). We reported results using small values of $k$: $[k = 2, k = 3, k = 4]$, to show that $k = 3$ is sufficient for direct video reconstruction. We will elaborate our parameter selection process in more detail in the camera-ready version.

## I.2   How To Choose $\delta$?

In **Appendix F.3 Adaptive Reference Latent Index Ablations**, we conducted quantitative experiments regarding different values (in the table, the rows correspond to different $k$ values, while the columns vary $\delta$). In that experiment, we report PSNR, SSIM, and LPIPS results. As a result of this experimental validation, we observe that the best PSNR, SSIM, and LPIPS scores are obtained when $\delta = 3$. Therefore, in all of our experiments in the main paper and supplementary videos, we use $\delta = 3$.

# J   Proposed method on Wan 2.1 (for Flow Matching models in general)

**K-RNR**, along with our dynamic view synthesis approach, is directly compatible with Wan 2.1 without requiring any modifications. Furthermore, in the section below, we illustrate how K-RNR

enables us to **bypass traditional iterative inversion schemes**, offering a more efficient, non-iterative alternative.

In Wan noise scheduler, $\epsilon' = \alpha_t x_0 + \sigma_t \epsilon$ operation is performed, where $x_0$ is the VAE-encoded latent and $\epsilon$ is sampled from a standard normal distribution.

Furthermore, $\alpha_t + \sigma_t = 1$. From now on, we will use $\sigma_t = (1 - \alpha_t)$ parameterization.

**We pose the following question:** *How effective is K-RNR when used without relying on any inversion process?*

To do so, we set $\epsilon^{(1)} = \epsilon \sim \mathcal{N}(0, I)$ and we followed our recursive noise representation formula:

**K-RNR in Flow Matching**

$$\epsilon^{(k)} = \alpha_t x_0 + (1 - \alpha_t)\epsilon^{(k-1)} \tag{1}$$

When this recursion is solved, we obtain a closed-form solution again in the form of:

$$\epsilon^{(k)} = \left[\sum_{i=1}^{k}(1 - \alpha_t)^{i-1}\alpha_t x_0\right] + (1 - \alpha_t)^k \epsilon \tag{2}$$

Importantly, $x_0$ is sampled from the Wan 3D-VAE using `argmax-sampling`, which uses the mode = mean of the latent distribution. Hence, $\mathbb{E}[x_0] = x_0$ and $\mathrm{VAR}[x_0] = 0$.

Now let's analyze the statistics and behavior of Eq. (2):

$$\mathbb{E}[\epsilon^{(k)}] = \left[\sum_{i=1}^{k}(1 - \alpha_t)^{i-1}\alpha_t \mathbb{E}[x_0]\right] = \left[\frac{1 - (1 - \alpha_t)^k}{\alpha_t}\right]\alpha_t \mathbb{E}[x_0] = \left[1 - (1 - \alpha_t)^k\right]\mathbb{E}[x_0]$$

$$\mathrm{VAR}[\epsilon^{(k)}] = (1 - \alpha_t)^{2k}$$

For the default setting, $\alpha_t = 0.07$.

**Behavior of the mean.** When $k = 1$, $\mathbb{E}[\epsilon^{(1)}] = 0.07\mathbb{E}[x_0]$. As $k \to \infty$, $\mathbb{E}[\epsilon^{(\infty)}] \to \mathbb{E}[x_0]$, so it gets $\frac{1}{0.07} \approx 15\times$ larger, hence **exploding**.

**Behavior of the variance.** Note that we did not use inverted latents for the $\epsilon^{(1)}$ but directly set it as standard normal, different from our paper setting. This results in a completely opposite behavior when it comes to variance. As $k \to \infty$, $\mathrm{VAR}[\epsilon^{(\infty)}] \to 0$, hence it is **vanishing**.

