# OpenReview forum: "Dynamic View Synthesis as an Inverse Problem"
_NeurIPS.cc/2025/Conference — NeurIPS 2025 poster_

### Official Review · Reviewer_zNp9 · 2025-06-05

**Clarity:** 2
**Significance:** 3
**Originality:** 3
**Rating:** 4
**Confidence:** 3

**Summary:**

This paper works on the problem of dynamic view synthesis. Different from existing works, it models this problem as an inverse problem. Specifically, it steers the pretrained video diffusion model to generate a video that matches the content that appears in the original video while hallucinating plausible content unseen in the original video. Compared to previous works, it does not require fine-tune the diffusion model.

**Questions:**

* My key question is regarding the motivation of K-RNR.
  * To tackle the Zero Terminal SNR problem, there are two strategies mentioned in the paper:
    1. Use the same zero terminal SNR noise schedule as the training time, but begin sampling at an earlier index with non-zero SNR (as mentioned in section 4.1)
    2. Use a positive terminal SNR as mentioned in section 4.2.
  * Does the proposed K-RNR belong to the second strategy?
  * In my understanding, in the second strategy, the noise schedule of training and inference (i.e., for novel view synthesis) is different. Will this difference affect the performance?
  * From the current paper, I cannot get a global understanding of the method. I suggest that the author add an algorithm figure to integrate the proposed K-RNR into the full pipeline. How does the K-RNR work with DDIM? Maybe I have some misunderstanding about the method, but I'm happy to change my mind if the author can clarify the technique details.
  * From the experimental results, K-RNR produces good results. Can the author provide more insights on the K-RNR?

**Ethical Concerns:**

["NO or VERY MINOR ethics concerns only"]

**Final Justification:**

All my concerns are addressed in the rebuttal. I think this paper is a valuable contribution to the community. I raise my score to weak accept.

**Quality:**

3

**Strengths And Weaknesses:**

### Strengths
* The motivation is clear. Modeling the dynamic view synthesis problem as the inverse problem is interesting.
* The results seem of higher quality than previous methods.

### Weaknesses
* The main idea is neat. But for me, the method part is hard to follow. See the Questions section.
* Figure 2 is not so clear.
  * I suggest that the author add a new column to show the wrapped target view. From the current version, it is not very easy for me to figure out that bcd is the inpainted version (or the synthesized target view) of a.
  * What is the meaning of strength? Is it the value of gamma in L129?
* In L176, the paper mentions "k=1 axis" but k is not defined before.
* I suggest the author add a comparison on runtime to compare their method with the previous methods that require fine-tuning, including the fine-tuning time and the inference time. This can help the readers better understand its benefits over the prior art

---

> ### Author Rebuttal · Authors · 2025-07-26
>
> # How to Set Terminal SNR During Inference?
>
> We would like to clarify any ambiguities regarding our treatment of zero-terminal SNR. To ensure our approach is fully understood, we provide additional motivation behind our strategy for handling zero-terminal SNR during inference.
>
> ---
>
> ## Motivation | How Not To Set Terminal SNR
>
> In this section, we focus on why the terminal SNR must not be set to $0$ in the K-RNR formulation.
>
> Let's start our motivation by analyzing K-RNR discrete formulation in Equation (1) for an extreme case: When  $t =T$ hence $\bar{a}_T=0$.
>
> For your convenience, we first start by rewriting the Equation (1) here:
>
> $$
> \left(\sum_{i=1}^k \sqrt{\bar{a}_t}(\sqrt{1 - \bar{a}_t})^{i-1}\right)x_0 + (\sqrt{1 - \bar{a}_t})^k\epsilon^{inv}  \tag{1}
> $$
>
> For $t =T$ when $a_T=0$, it becomes:
> $$
> \left(\sum_{i=1}^k \sqrt{\bar{a}_T}(\sqrt{1 - \bar{a}_T})^{i-1}\right)x_0 + (\sqrt{1 - \bar{a}_T})^k\epsilon^{inv} \tag{2}
> $$
>
> $$
> = \left(\sum_{i=1}^k 0 \times \sqrt{1}\right)x_0 + (\sqrt{1})^k\epsilon^{inv}
> $$
>
> $$
> = \epsilon^{inv}
> $$
> proving that, when K-RNR is used in conjunction with zero-terminal SNR noise scheduler in the inference. In conclusion, for K-RNR to get high fidelity reconstructions, the first term in Equation (1) must be non-zero:
>
> $$
>  \left\| \left(\sum_{i=1}^k \sqrt{\bar{a}_t}(\sqrt{1 - a_t})^{i-1}\right)x_0\right\| > 0 \tag{3}
> $$
>
> which is possible when terminal SNR(t) = $\frac{\bar{a}_t}{1-\bar{a}_t} > 0$. This conclusion justifies again why we must use positive-terminal SNR noise scheduler in the inference.
>
> ---
>
> ## Technical Details | Then How To Set Terminal SNR
> **Q: Which strategy does K-RNR belong to?**
>
> As outlined by the reviewer, there can be 2 natural approaches to obtain positive-terminal SNR given a video model trained with zero-terminal SNR. In a 50 timesteps inference setting, these approaches can be summarized as follows.
>
> ### 1. Straightforward Approach | Terminal Timestep Shifting
> * In this strategy, one can shift the terminal timestep $T=999$ to an earlier timestep $T'$ where $T > T' > T-1$. In 50 timestep inference setting with uniform noise scheduler $T-1=979$. In timestep shifting strategy, **as the diffusion path is not shortened** (still 50 timesteps), the proposed framework has a lot of flexibility to both reconstruct the original layout + details and inpainting the occluded regions.
>
> **Q: Is modifying the noise scheduler affects performance?**
>
> * Timestep shifting is a valid approach as the diffusion models are trained with 1000 timesteps and know how much noise to remove in each of these 1000 timesteps. Importantly, timestep shifting approach is also adapted by the modern Diffusion DiT models and it is configurable by the users.
> * In our  experiments, only `Zero Terminal SNR Collapse` subsection under `Motivations` section in supplementary videos uses this strategy.
>
> ### 2. Challenging Approach | Shortening Diffusion Steps with `strength` parameter
>
> **Q: What is the meaning of `strength` parameter in diffusion?**
>
> The **`strength`** parameter is a inference step controlling mechanism in diffusion models which is set a ratio value between $0.0$ and $1.0$. Given a 50 timestep inference, if **`strength`** ratio is set to $1.0$, that means inference will start at terminal timestep $T=999$ and it will run for 50 timesteps: $[999, 979, 959, 939, 929 ..., 29, 19]$. When **`strength`** ratio is set to  $0.95$ the inference runs for $floor(50\times0.95)=47$ timesteps: $[939, 929 ..., 29, 19]$.
>
> * In this strategy, the timesteps are chosen in SDEdit style with strength parameter is set to values less than 1 such as 0.95.
> * The crucial difference from this approach to terminal timestep shifting approach is that, in this strategy the **denoising path is shortened** accordingly to the chosen strength parameter.
> * This strategy sets a more challenging dynamic view synthesis environment as the video diffusion model now has less flexibility to reconstruct in high fidelity and inpaint the occluded regions.
> * This situation is demonstrated in Supplementary Videos `Approaches to Zero Terminal SNR Collapse Problem`  section with $Strength=0.95$ and $Strength=0.88$ cases.
> * In these experiments, it can be seen that:
>     * Strength=$0.88$ shortens the path more aggressively yielding results with the reconstruction of occluded regions.
>     * Strength=$0.95$ has the flexibility to inpaint a portion of the occluded region with the expense of identity drift of the monkey.
>
> ### Our Approach | Hypothesis and Validity
> * As explained above, we experimented with both strategy on our end. With that, as explicitly mentioned in the `Experiments` section Line 262, we report all the results in the paper and supplementary videos except the ones under `Zero Terminal SNR Collapse` subsection (which belongs to the **first strategy**) using the **second strategy**: `Challenging Approach | Shortening Diffusion Steps ` with terminal timestep = $0.95T$ in which $\sqrt{\bar{a}_t} = 0.034$
> * Our hypothesis was that a better initialized noise should be able to correct the identity drift issue experienced by the baseline, furthermore it should be capable of fully completing the occluded regions plausibly.
> * Our 100+ experiments in the supplementary videos show that K-RNR accounts for original scene fidelity while SLM accounts for the propagation of the content aware information to occluded regions.
>
> ---
>
> # Full Pipeline | Global Understanding of the Method
>
> **Q: From the current paper, I cannot get a global understanding of the method. I suggest that the author add an algorithm figure to integrate the proposed K-RNR into the full pipeline.**
>
> In this section, we would like to elaborate more on global understanding of the methodology by explaining our algorithm step by step.
>
> 1. Given a source video, we first apply **camera transformation**.
>     * Each frame is lifted up to 3D point cloud using depth information.
>     * Desired camera transformation is applied in point cloud space.
>     * Camera transformation applied point clouds lifted down to frame space.
>     * The resulting camera transformation applied video is called **`render`** throughout the paper and supplementary videos.
>     * Render video consists of occluded regions due to camera transformation as shown in supplementary videos.
>     * We also extract the visibility mask in this stage. Visible regions in render video are represented as 1 and occluded regions are represented as 0.
> 2. Encoding render video through 3D-VAE.
>     * After VAE encoding we get the latent $x_0$.
>     * We also map depth information and binary visibility mask to CogVideoX's latent space.
> 3. Applying **DDIM Inversion** to render video.
>     * A common recipe for inverting a video for high quality reconstruction in CogVideoX-5B  is using **250 forward steps inversion** [1].
>     * However, as our K-RNR accounts for high quality fidelity we only use **20 forward steps** inversion in the inversion.
>     * As a result of inversion, we obtain inverted latent $\epsilon^{inv}$
> 4. Stochastic Latent Modulation (SLM)
>     * Based on the visibility mask and depth information, we get stochastic replacement indices for randomly replacing occluded latents.
>     * We replace occluded regions of $x_0$ based on the selected indices.
>     * We replace occluded regions of $\epsilon^{inv}$ with the same selected indices.
>     * **Potential Question |** If the already existed visible scene features are utilized for occluded regions, how is SLM inpaints diverse, context-aware regions?
>         * It is because of the positional embeddings. CogVideoX uses 3D RoPE. Although already existing features are re-used for the occluded regions in SLM procedure these regions are rotated differently.
>         * RoPE rotates low-dimensional 2D feature chunks faster than high-dimensional ones, preserving semantics in higher dimensions and making SLM a context-aware inpainting method.
> 5. K-order Recursive Noise Representation (K-RNR)
>     * After performing latent inpainting to $x_0$ and $\epsilon^{inv}$, we perform Equation (1):
>     * $$ \epsilon^{(k)} = \left(\sum_{i=1}^k \sqrt{\bar{a}_t}(\sqrt{1 - \bar{a}_t})^{i-1}\right)x_0 + (\sqrt{1 - \bar{a}_t})^k\epsilon^{inv} $$
>     * Which is followed by Adaptive Normalization:
>     * $$ \epsilon^{(k)} = AdaIN(\epsilon^{(k)}, \epsilon^{(\delta)})$$
> 6. Video Diffusion Model
>     * The resulting $\epsilon^{(k)}$ is used as initial noise.
>     * Resulting video with desired camera transformation is obtained without any modification to backbone.
> ```
> [1] Yatim, Danah, Rafail Fridman, Omer Bar-Tal, and Tali Dekel. "DynVFX: Augmenting Real Videos with Dynamic Content." arXiv preprint arXiv:2502.03621 (2025). | Appendix A.2.Keys and Values Extraction section.
> ```
> ---
>
> ## Paper Writing | Figure 2 Details
> **Q: It is not very easy for me to figure out that bcd is the inpainted version (or the synthesized target view) of a.**
>
> The reviewer is correct: Figures (b), (c), (d) are inpainted versions of (a) with different noise initialization strategies. We thank the reviewer for the suggestion and will ensure greater clarity in the camera-ready version.
>
> ---
>
> ## Paper Writing | $k=1$ axis statement
> **Q: The paper mentions $k=1$ axis but k is not defined before.**
>
> We used the term "$k = 1$ axis" without defining it, as the graph already starts at $k = 1$, our initial condition. We thank the reviewer and will clarify this in the camera-ready version.
>
> ---
>
> ## Paper Writing | Adding a runtime comparison
> **Q: I suggest the author add a comparison on runtime to compare their method with the previous methods that require fine-tuning.**
>
> We thank the reviewer for this very good suggestion, which we believe will strengthen our paper. The additional latency introduced by K-RNR + SLM on top of CogVideoX’s inference is **less than a second** on a single L40 GPU which is significantly lower than prior methods that require fine-tuning. We will include this runtime comparison in the revised version.

---

> > ### Comment · Reviewer_zNp9 · 2025-08-04
> >
> > Thanks for the rebuttal. My main concerns have been addressed. I'm now lean positive to this paper and will raise the score.

---

> > > ### Author Response · Authors · 2025-08-04
> > >
> > > Dear Reviewer,
> > >
> > > We would like to thank you for your acknowledgment of our clarifications. We sincerely appreciate your willingness to raise your score. We will incorporate all of the clarifications discussed in our rebuttal into the final manuscript.
> > >
> > > Best regards, Authors

---

> > > ### Author Response · Authors · 2025-08-08
> > >
> > > Dear reviewer,
> > >
> > > We are glad that you to hear about your positive comment `"My main concerns have been addressed. I'm now lean positive to this paper and will raise the score."`. As we are coming to the end of the rebuttal period today, we kindly wanted to send a reminder about this.
> > >
> > > Best,
> > >
> > > Authors

---

### Official Review · Reviewer_U62z · 2025-06-30

**Clarity:** 3
**Significance:** 2
**Originality:** 2
**Rating:** 4
**Confidence:** 5

**Summary:**

This paper explores how to leverage a pretrained video diffusion model to perform novel view synthesis of dynamic scenes from monocular video input. It investigates the issue of zero-terminal SNR collapse and proposes two solutions: K-order Recursive Noise Representation (K-RNR), which preserves seen regions through effective noise initialization, and Stochastic Latent Modulation (SLM), which inpaints newly revealed regions. The work demonstrates the versatility of diffusion models in dynamic view synthesis tasks.

**Questions:**

I have the following concerns regarding this work, listed in order of urgency.

1. **Missing relevant literature**
   - This paper lacks sufficient discussion of related works that explore the versatile applications of diffusion models, such as the direct use of pretrained models for inpainting without fine-tuning. Although the paper builds on similar concepts, it does not adequately cite or discuss with relevant prior works, such as [1, 2].
   - Section 4.3 of this paper bears a strong resemblance to prior work [3] presented at CVPR 2023; however, the paper does not adequately cite or discuss its relationship with [3].

2. **Why not simply fix the seen regions, as done in [3]?** To make the initialization more consistent with $x_0$, this paper proposes K-RNR. Similarly, [3] suggests directly fixing the seen regions. The authors should justify why K-RNR offers advantages over the simple masking approach used in [3], either through theoretical explanation or experimental validation.

3. **Overclaim.** The paper claims that their method can produce results consistent with the rules of the physical world (line 241). However, I believe this statement is overclaimed. There is no guarantee that the generated results strictly adhere to physical laws; at best, they ensure visual plausibility rather than physical accuracy.

4. **I could not find relevant experiments analyzing the sensitivity of the concrete values of $k$ and $\delta$**. How sensitive is the performance to the choice of these parameters? Do small changes in their values significantly affect the results, or is the method robust and largely insensitive across different datasets?

5. **You still use a strength of 0.95 to ensure $\bar{\alpha}_T>0$ (line 262).** However, K-RNR is theoretically designed to overcome this limitation. Therefore, why not set $\bar{\alpha}_T=0$, given that K-RNR already injects information from $x_0$ into the noise? Furthermore, you discuss the drawbacks of using $\bar{\alpha}_T>0$ in lines 161~166. Wouldn't these limitations also apply to your method, given that you still rely on $\bar{\alpha}_T>0$ for noise initialization?

6. **SLM Validity.** Why does randomly permuting latents in **S** into occluded regions in **M** produce coherent content? It would be valuable to see empirical comparisons between the proposed strategy and established baselines, including zero-filling, noise-filling, and content-aware approaches based on learning-free texture transfer and synthesis [4].

7. **The idea presented in Proposition 4.1 appears to be intuitively straightforward.** The authors could simplify this section to shorten the presentation.

8. **Typo:** At the bottom of page 8, the caption of the table incorrectly begins with "Figure 9", but the referenced content is a table, not a figure.



[1] RePaint: Inpainting using Denoising Diffusion Probabilistic Models

[2] SDEdit: Guided Image Synthesis and Editing with Stochastic Differential Equations

[3] RGBD2: Generative Scene Synthesis via Incremental View Inpainting using RGBD Diffusion Models

[4] GIMP Resynthesizer Plugin Suite. https://github.com/bootchk/resynthesizer

**Ethical Concerns:**

["NO or VERY MINOR ethics concerns only"]

**Final Justification:**

The authors have effectively addressed all of my concerns. I believe this paper will contribute positively to the field.

**Limitations:**

While the paper acknowledges several limitations, it also appears that their approach relies on access to known camera trajectories in order to generate novel video content. This is challenging when camera parameters are either unknown or time-consuming to obtain.

**Paper Formatting Concerns:**

The contribution paragraphs are separated by unnecessarily large spacing.

**Quality:**

2

**Strengths And Weaknesses:**

**Strength**: Their method is straightforward and relatively easy to understand.

**Weakness**: Although this paper was not difficult for me to read, its writing could be simplified to make it more accessible to a broader audience. For instance, the authors could use simpler terminology and provide more detailed explanations. Additionally, the experimental section lacks sufficient persuasiveness. For further limitations, please refer to the `Questions` section described later.

---

> ### Author Rebuttal · Authors · 2025-07-26
>
> ## Missing Ablations | We kindly note that the requested ablations are already performed.
> **Q4: I could not find relevant experiments analyzing the sensitivity of the concrete values of $k$ and $\delta$**
>
> We kindly want to inform the reviewer that we have already conducted these experiments both quantitatively and qualitatively.
>
> **Ablations of $k$ values |** In `Appendix F.2: Discrete K-order Ablations` section, we conducted the **quantitative** experiments regarding different $k$ values. We report PSNR, SSIM, LPIPS results. Moreover, we comment on the experimental results and its validity in the same section F.2. Furthermore, in our supplementary material where we share 100+ video results, the reviewer can find **qualitative** results with different $k$ values in almost **all experiments**.
>
> **Ablations of $\delta$ values |** In `Appendix F.3 Adaptive Reference Latent Index $\delta$ Ablations`, we  conducted the quantitative experiments regarding different $\delta$ values (in the table, the rows has typo where $k$ =... needs to be $\delta$ = ...). Here, we also report PSNR, SSIM, LPIPS results.  In all of our experiments in the paper and supplementary videos, we use $\delta$ = 3. The reason for this choice is also elaborated in this section **with experimental validity**.
>
> ---
>
> ## Missing Experiments | We kindly note that the requested experiment is already performed.
> **Q2: Why not simply fix the seen regions?**
>
> We kindly want to inform the reviewer that we have already conducted this experiment.
>
> Indeed, K-RNR with $k=1$ in conjunction with SLM setting directly corresponds to using original DDIM inverted latent for the seen regions and inpainting the occluded regions:
> $$
>  \text{For K=1:} \quad \left(\sum_{i=1}^{k=1} \sqrt{\bar{a}_T}(\sqrt{1 - a_T})^{i-1}\right)x_0 + (\sqrt{1 - a_T})^k\epsilon^{inv} \tag{1}
> $$
> $$
>  = \sqrt{\bar{a}_T}x_0 + (\sqrt{1 - a_T})\epsilon^{inv} \tag{2}
> $$
> Reviewer's suggested experiment setting is comprehensively experimented qualitatively and quantitatively. In Supplementary Videos section `Noise Initialization Ablations` where we share **12 noise initialization** methods, we allocated **7 of them for this setting**: The reviewer can see clearly the evolution of the resulting dynamic views from standard DDIM Inversion to Reviewer's desired setting K-RNR with $k=1$, from recursive K-RNR starting from $k=2$ to $k=6$. The evolution of the dynamic views are very clear in these experiments.
>
> **Clarifying a potential misunderstanding:** We believe the nature of the zero-terminal SNR issue may be misunderstood, therefore we kindly provide a detailed clarification step by step:
>
> * The reviewer suggests to compare our method with [3] which fixes the seen regions using masked attention.
> * However, we kindly note that there is no need to conduct this experiment to see that it won't work.
> * The reason is that, for [3] to work in CogVideoX which is trained with zero-terminal SNR, the reviewer assumes DDIM inverted latent, when used as the initial noise directly, reconstructs the original scene in a meaningful manner.
> * Based on above assumption, it is suggested that applying masked attention to preserve the seen regions would lead to perfect reconstruction in those seen areas.
> * **However, initial assumption doesn't hold true.**  Indeed, one of the main points of our paper is showing that with zero terminal SNR the injectivity property is broken down and DDIM inverted latents reconstructs washed out results.
>     * We also kindly point out the following sections in `website.html` supporting our observations:
>         * `Zero Terminal SNR Collapse` section
>         * `Initial Noise Ablations` section
>
> ---
>
> ## Technical Elaborations | Why not set $\bar{a}_T=0$?
> **Q5: You still use a strength of 0.95 to ensure $\bar{a}_T > 0$, why not set $\bar{a}_T=0$?**
>
> When $\bar{a}_T = 0$, Equation (1) reduces to standard DDIM inversion which is the entire point of us for writing this paper: To demonstrate standard DDIM inversion does not work under zero terminal SNR setting.
>
> Let's see this situation step by step. in Equation (1), when $t = T$  where $\bar{a}_T = 0$, as the reviewer proposed:
>
> $$
> \left(\sum_{i=1}^k \sqrt{\bar{a}_T}(\sqrt{1 - \bar{a}_T})^{i-1}\right)x_0 + (\sqrt{1 - \bar{a}_T})^k\epsilon^{inv}
> $$
>
> $$
> = \left(\sum_{i=1}^k 0 \times \sqrt{1}\right)x_0 + (\sqrt{1})^k\epsilon^{inv}
> $$
> $$
>  = \epsilon^{inv}
> $$
> Clearly, when $t=T$ ; $\bar{a}_T = 0$ the formulation reconstructs the standard DDIM inversion which is already shown to be problematic to directly use in regard of **norm deviation** in `Figure 5`, with its **washed out results** qualitatively shown in `Motivations`, `Approach to Zero Terminal SNR Collapse Problem`, `Noise Initialization Ablations` sections of the Supplementary Videos and **quantitatively** shown in `Figure 9`.
>
> The entire point of K-RNR is setting VAE encoded video latent $x_0$ as the pivot latent (meaning it is fixed) and updating the standard DDIM inverted latent in our new noise formulation K-RNR as described in Line 182 in the main paper. To share information from $x_0$ and obtain a better aligned initial noise as shown in Cosine Similarity graph in Figure 4.a, we set  $t = 0.95T$ which corresponds to $a_t$ = 0.0012 in our experiments.
>
> **Wouldn't these limitations also apply to your method?** We kindly point out that we already conducted the ablation of this concern. Please refer to Supplementary Videos `Approaches to Zero Terminal SNR Collapse Problem` section to see comparison.
>
> **Additional Question |** If the reviewer's question is would K-RNR still work without shortening the diffusion path, the answer is **yes.** All of the K-RNR experiments in `Zero Terminal SNR Collapse` section in supplementary videos are conducted with $strength=1.0$ and shifting the terminal SNR. Please refer to our answer to **Reviewer zNp9** with the section name  `Easier Approach | Terminal Timestep Shifting`
>
> ---
>
> ## Technical Elaborations | SLM Validity
> **Q6: Why does randomly permuting latents in S into occluded regions in M produce coherent content?**
> * It is because of the positional embeddings. CogVideoX utilizes 3D Rotary Positional Embeddings (RoPE). Although already existing features are re-used for the occluded regions in SLM procedure, after patchification, these regions are rotated differently.
> * Moreover, due to the nature of RoPE, each 2-dimensional chunk of total D dimensional feature are also rotated in different frequencies. While 2-dimensional chunks in lower dimensions are rotated faster, higher dimensions are rotated slowly.
> * This means that, while lower dimensions of replaced occluded regions are being affected more by the position information, the higher dimensions are still holding the semantic information (as they rotated slower and less effected by the position information). Hence, SLM is a context aware inpainting mechanism.
>
> **Requested SLM Ablations.**
>
> | Method        | PSNR (↑) | SSIM (↑) | LPIPS (↓) |
> |---------------|----------|-----------|----------|
> | Zero Filling  |     9.16     |    0.317       |     0.911     |
> | Noise Filling |     10.07     |  0.356         |     0.906     |
> | SLM           |    24.12      |  0.873         |    0.086      |
>
> We want to inform the reviewer that GIMP is not comparable with SLM. SLM is a purely latent space operation with channels $C=16$. GIMP, on the other hand, completely works in the image space with $C=3$
>
> ---
>
> ## Paper Writing | Shortening section 4.1
> **Q7: The idea presented in Proposition 4.1 appears to be intuitively straightforward.**
>
> We thank reviewer for suggesting shorten this section. However, this section is at the heart of our method motivation.  The purpose of this section is to motivate the underlying inversion problem and demonstrating readers why $\sqrt{\bar{a}_T} = 0$  shouldn't be used in our K-RNR formulation. Given that there still seems to be a confusion about why we must not set  $\sqrt{\bar{a}_T}=0$ for high fidelity reconstruction, we will make sure to elaborate more on this section in our camera ready version.
>
> ---
>
> ## Paper Writing | "The rules of the physical world" statement
> **Q3: Overclaim.**
>
> We used this statement in Line 241 to highlight that although physically non-plausible and non-dynamic frames are given to VAE encoder as shown in Figure 7.b, through our K-RNR formulation where $\epsilon^{inv}$ comes from the non-filled, occluded render frames, it can generate physically plausible (we stated this way in Figure 7 caption) videos. We absolutely agree on this with the reviewer and will change the statement from "The rules of the physical world" to "physically plausible" as we mentioned in the Figure 7 caption.
>
> ---
>
> ## Paper Writing | Literature Review
> **Q1: Missing relative literature.**
>
> **Strong resemblance of section 4.3 to prior work |**  Indeed, section 4.3 starts with the statement "Following prior works [58, 60, 20, 59, 55, 15] ..." which clearly suggests that we do not propose any novel camera conditioning mechanism in this paper but rather utilize the existing, well-established mechanism by the prior works. Currently 2 ways of conditioning the camera information exists in dynamic view synthesis with video diffusion priors methods.
>
> 1. Lifting each frame to 3D point cloud space, applying camera transformation and lifting down to frame space as done in:
>     * `[58] TrajectoryCrafter`
>     * `[60] Recapture`
>     * `[20] Reangle-a-video`
>     * `[59] Viewcrafter`
>     * `[55] TrajectoryAttention`
>     * `[15] Diffusion as Shader`
> 2. Training a camera condition embedder as done in:
>     * `[2] ReCamMaster`
>
> Being a **training-free** methodology we preferred to use first camera conditioning mechanism. We thank reviewer for pointing out the original paper introducing this camera conditioning mechanism. We will make sure to cite this paper on camera-ready version.
>
> **Citing SDEdit and RePaint |** We will make sure to cite these works in camera-ready version.

---

> > ### Comment · Area_Chair_wYsv · 2025-08-01
> > **Please respond**
> >
> > Dear reviewer,
> >
> > the authors seem to have addressed your concerns. Can you please read the rebuttal and respond to it asap?

---

> > ### Comment · Reviewer_U62z · 2025-08-04
> > **Good Rebuttal Response**
> >
> > The authors have effectively addressed my concerns, and I appreciate their efforts. I acknowledge that I overlooked certain aspects of their submission, and I would be happy to raise my rating if the authors could incorporate these changes into their final camera-ready version. Thank you for their thorough clarifications.

---

> > > ### Author Response · Authors · 2025-08-04
> > >
> > > Dear Reviewer,
> > >
> > > We would like to thank you for your thoughtful consideration and acknowledgment of our clarifications. We sincerely appreciate your willingness to reconsider your evaluation. We will incorporate all of the clarifications discussed in our rebuttal into the final manuscript.
> > >
> > > Thank you again for your valuable feedback and support.
> > >
> > > Best regards,
> > > Authors

---

### Official Review · Reviewer_gB27 · 2025-06-30

**Clarity:** 3
**Significance:** 3
**Originality:** 3
**Rating:** 4
**Confidence:** 4

**Summary:**

This paper addresses the challenges of dynamic view synthesis from monocular videos by introducing a training-free framework. The paper first identifies zero terminal signal-to-noise collapse problem that impedes deterministic inversion in models trained with zero terminal SNR schedules and proposes K-RNR. To further resolve the inverse problem of synthesizing newly visible regions occluded from the initial camera trajectory, the authors introduce Stochastic Latent Modulation as a mechanism to complete occluded latent regions using contextual permutations.

**Questions:**

1. How robust is the framework to hyperparameters $ \delta $ and $ k $ in adaptive K-RNR? The results seem sensitive to the configuration of the adaptive component and blows up as $ k $ is increases past a small threshold. Since larger $ k $ accumulates $ x_0 $ in the weighted sum, it makes sense the values grow uncontrollably. Have you considered a weighted average to keep the norm more controlled?
2. Have you tried approaches other than random permutation of the visible background for stochastic latent modulation? For instance, repeating the closest latent neighbor is spatially smoother than random permutations. I speculate redistributing background patches may tradeoff diversity of the generation for large occlusions especially and degrade quality as well since the latents are out-of-distribution. Initializing with pure random noise for the entirely unseen regions seem more ideal.
3. Since the framework is training-free and based on point cloud warping, when there are large deviations from the initial camera trajectory and thus large regions become newly visible, the method can potentially produce unstable or incoherent results. How robust is the framework to challenging scenarios?

**Ethical Concerns:**

["NO or VERY MINOR ethics concerns only"]

**Final Justification:**

The authors have provided missing ablations and discussions that clarifies the method design and effects of hyperparameter changes. They have additionally added more context to explicit camera conditioning mechanism which was largely missing initially. Although the proposed method has novelty and experiments justify the improvements over baselines, I would like to maintain my borderline acceptance rating due to inherent limitations of training-free and point cloud warping approach. When large occluded regions become visible or errors in 3D point cloud estimation accumulate, the method produces artifacts with unstable results. These challenging cases are more practical scenarios with impactful applications.

**Limitations:**

Yes.

**Paper Formatting Concerns:**

No major formatting issues.

**Quality:**

3

**Strengths And Weaknesses:**

**Strengths**
1. The figures are informative and comprehensive in presenting the effects of K-order noise and the ablation identifies the contribution of each component well.
2. The underlying problem is identified and motivated clearly with concrete formalizations.
3. Experiments are conducted across multiple datasets and baselines, validating the performance of the proposed method.

**Weaknesses**

1. Explicit camera conditioning section needs more elaboration. How are the camera intrinsics of videos estimated? Currently the depth maps are predicted per frame independently over the video which would result in inconsistent scales across local point clouds. How are the local point clouds registered into a consistent global point cloud? How are camera poses computed and aligned to a unified scale?
2. Correction to CogVideoX 3D VAE latent shape. On L118, the final output shape of the latent should be $ (1 + \lceil \frac{F-1}{4} \rceil) \times 2C \times \frac{H}{8}  \times \frac{W}{8}$. The first conditioning frame is encoded independently and the rest are temporally compressed by a factor of 4.

---

> ### Author Rebuttal · Authors · 2025-07-27
>
> ## How robust is the framework to hyperparameters $\delta$ and $k$ in adaptive K-RNR?
> **Q: How robust is the framework to hyperparameters $k$ and $\delta$ in adaptive K-RNR?**
> * We thank the reviewer for this insightful question. As stated in the experiments section, we fixed $k=10$ and $\delta=3$ in all quantitative experiments which yields the state-of-the art performance.
> * **Why $\delta=3$:**  In `Appendix F.3 Adaptive Reference Latent Index Ablations`, we conducted the quantitative experiments regarding different  values (in the table, the rows has typo where  $k$=... needs to be  $\delta$= ...). As a result of this experiment, we see that the best scores are obtained when $\delta=3$ and in all of our experiments in the paper and supplementary videos, we use $\delta= 3$.
> * **How to choose $k$?** Indeed, we obtained the best results when we set $k=6$ and $k=10$. We don't tweak the $k$ value per video-camera pair. Here we want to clarify an important ambiguity:
>
>     * In the **monkey** example, we illustrate how increasing $k$ in K-RNR affects the rendered video. The goal is to help readers observe how video content evolves with different $k$ values. As noted, $k = 6$ yields plausible results.
>
>     * In the **elephant** and **duck** examples, we demonstrate K-RNR’s effectiveness in reconstructing source videos without occlusion (and thus without SLM). We report results for small $k$ values ($k = 2$, $k = 3$, $k = 4$), showing that $k = 3$ is sufficient for direct reconstruction.
>
>     * Additionally, in `Appendix F.2: Discrete K-order Ablations`, we provide quantitative results across different $k$ values.
>
>
> ---
>
> ## Weighted average of $x_0$ and $\epsilon^{(k)}$
> **Q: Have you considered a weighted average to keep the norm more controlled?**
>
> Indeed, proposing weighted average again introduces 2 parameters which may be further required to tune for each iteration in total of $k$ iterations in Equation (1). On the other hand, as discussed above, our method is quite robust when $k=10$ (or $k=6$) and $\delta=3$ are used.
>
> ---
>
> ## Details on external camera conditioning
> **Q: How are the camera intrinsics of videos estimated?**
>
> Our approach employs a **fixed camera model** rather than estimating intrinsics directly from input videos, which is a common and robust practice for video-based 3D applications when camera calibration is unavailable.
>
> Following `TrajectoryCrafter`, we constructed the camera intrinsic matrix using **hardcoded parameters** optimized for CogVideoX video resolutions. This configuration assumes a perspective camera model with equal focal lengths in both $x$ and $y$ directions and centers the principal point appropriately for the input video dimensions.
>
> **Q: How are the local point clouds registered into a consistent global point cloud?**
>
> Indeed, we do not apply any additional registration operation for obtaining global point cloud as we already use temporally consistent depth maps extracted from `DepthCrafter`. `DepthCrafter` processes the entire video sequence as a unified input, producing depth maps that are inherently consistent across frames.
>
> It is important to note that at the time of our submission, more advanced video depth models such as `Video Depth Anything` [1] published in CVPR 2025 were available and outperformed `DepthCrafter`. However, to ensure a fair comparison with TrajectoryCrafter, we consistently used `DepthCrafter` in all quantitative and qualitative experiments.
>
> ```
> [1] Chen, Sili, et al. "Video depth anything: Consistent depth estimation for super-long videos." Proceedings of the Computer Vision and Pattern Recognition Conference. 2025.
> ```
> **Q: How are camera poses computed?**
>
> The camera poses are computed in a **spherical coordinate system** where the radius parameter is automatically scaled based on the scene's depth characteristics. More specifically, we extract the depth value at the `central pixel` of the reference frame and use this as a basis for determining appropriate viewing distances. This ensures that camera trajectories are scaled proportionally to the scene geometry, maintaining consistent viewing perspectives regardless of the absolute depth values.
>
> ---
>
> ## Point Cloud Warping Limitations
> **Q: Since the framework is training-free and based on point cloud warping, when there are large deviations from the initial camera trajectory and thus large regions become newly visible, the method can potentially produce unstable or incoherent results.**
>
> The reviewer is absolutely right. Indeed, as we acknowledge in our **Limitations and Broader Impact** section, stochastic latent modulation can also produce unstable or incoherent results when large regions become newly visible. To elaborate this point, we kindly note that we also qualitatively demonstrated how errors in the 3D point cloud warping can result in unpleasant results in `3D Point Cloud Rendering Artifacts` section of our supplementary videos.
>
> **Question by the Authors |** Given this limitation, why did we still choose using 3D point cloud warping as our camera conditioning mechanism?
>
> Currently, there are two dominant ways of  camera information conditioning in the dynamic view synthesis field among methods leveraging **video diffusion priors**:
>
> * **3D point Cloud Warping:** `TrajectoryCrafter`, `Recapture`, `Reangle-a-Video`, `Viewcrafter`, `TrajectoryAttention`, `Diffusion as Shader`
>
> * **Training a camera condition embedder** `ReCamMaster`
>
> Since our method is training free, and the additional latency introduced by K-RNR + SLM on top of CogVideoX's own inference is **less than a second** on a single L40 GPU, we chose the training-free camera conditioning mechanism.
>
> **Q:  How robust is the framework to challenging scenarios?**
>
> Our framework demonstrates strong robustness across a variety of challenging scenarios. Unlike our baselines, we evaluate our method on three diverse datasets, each targeting a different robustness aspect:
>
> * **OpenVid:** This dataset includes videos of human actions involving fine-grained, local motions such as gestures and hand movements. These are particularly difficult to preserve, and our model performs well in maintaining such subtle details.
>
> * **DAVIS:**  We use this dataset to assess the robustness of our framework in the presence of significant object and camera motion.
>
> * **Sora videos:**  Beyond real-world datasets, we also test our model on source videos generated by models like Sora to evaluate robustness to synthetic content.
>
> These evaluations collectively demonstrate that our framework generalizes well and maintains high-quality performance even under challenging or out-of-distribution conditions.
>
> ---
>
> ## Stochastic Latent Modulation vs other filling strategies
>
> **Q: Repeating the closest latent neighbor is spatially smoother than random permutations.**
>
> This is an excellent observation. In fact, during our experimentation phase, the closest latent neighbor approach was the first filling strategy we explored. As the reviewer correctly notes, this method often produces smooth and visually coherent results. However, despite its strengths in generating temporally consistent outputs, we observed that in certain cases it leads to context-unaware generations.
>
> To be concrete, in the **"cat wearing thick round glasses sits on a crimson velvet armchair"** source video, when we apply horizontal arc camera transformation the render video has occluded regions  behind the cat on the armchair. Using the `closest latent neighbor approach`, we consistently observed that these occluded regions were not filled with plausible continuations of the armchair. Instead, the method incorrectly filled them with textures from the background library, as if the occluded areas belonged to the library. This type of physically inconsistent hallucination was the issue that motivated us to explore alternative filling strategies and finalizing our method with a depth and context-aware stochastic latent modulation.
>
> **Q: Initializing with pure random noise for the entirely unseen regions seem more ideal.**
>
> Although it may initially appear that filling occluded regions with pure random noise is the simplest and more correct way, we kindly note that **this is not the case in reality**.
>
> Note that we first apply `DDIM Inversion` to obtain $\epsilon^{inv}$, followed by our `Occlusion Filling (SLM)` step, and finally apply `K-RNR`. As a result of this transformation sequence, the proposed `Occlusion Filling` strategy must inpaint the occluded regions at the same scale as the visible ones to ensure latent consistency.
>
> However, it is important to emphasize that the DDIM-inverted latent $\epsilon^{inv}$ is deterministic and does not follow a normal distribution (this situation is shown in our **Expected Norm Deviation** graph in Figure 5), since the stochasticity parameter $\sigma = 0$ during DDIM inversion. Consequently, when we fill the occluded regions in $\epsilon^{inv}$ with noise sampled from a normal distribution which differs in scale the resulting latent becomes out-of-distribution. This mismatch results in washed-out results again.
>
> To match the above theory with experimental results, we conducted quantitative comparison between `noise filling` and `SLM` on 25 OpenVid examples. The results are shown below.
>
> | Method        | PSNR (↑) | SSIM (↑) | LPIPS (↓) |
> |---------------|----------|-----------|----------|
> | Noise Filling |     10.07     |  0.356         |     0.906     |
> | SLM           |    24.12      |  0.873         |    0.086      |
>
> ---
>
> ## Correction to 3D VAE latent shape
> **Q:  On L118, the final output shape of the latent should be** (1 + $\lceil \frac{F-1}{4}\rceil \times 2C \times \frac{H}{8} \times \frac{W}{8}$)
>
> We thank reviewer for pointing this out. We will make sure to correct the latent shape in the camera ready version.

---

> > ### Comment · Reviewer_gB27 · 2025-08-05
> >
> > Thank you for providing in-depth discussions and clarifying my concerns. I still have confusions regarding the point cloud construction. Does DepthCrafter estimate depth maps with respect to a single frame of reference? Otherwise, registration would still have to occur with dynamic point clouds segmented to disentangle scene motion from camera motion for robustness. Also recent work [1] that estimates camera intrinsics as well as dynamic point clouds could help address the fixed camera model assumption. Although I do understand the techniques behind external camera conditioning are not this work's main contribution and are inspired by prior works, more discussion would improve comprehensiveness.
> >
> > ```[1] Zhang, Junyi, et al. "MonST3R: A Simple Approach for Estimating Geometry in the Presence of Motion." ICLR. 2025.```

---

> > > ### Author Response · Authors · 2025-08-05
> > >
> > > Thank you for raising this question. To clarify:
> > >
> > > DepthCrafter produces depth sequences in a **globally consistent relative scale** across the entire video, rather than applying independent normalization to each frame. In practice, the model predicts affine‐invariant depth values with a single, shared scale and shift for all frames in the sequence. Although the estimated depth is relative (not absolute metric depth), this shared scale/shift ensures that the resulting point cloud is constructed within a single, coherent reference frame for the entire video segment.
> > >
> > > We appreciate your thoughtful feedback and your willingness to help us further improve our work. We will review the attached recent paper with great interest and ensure that these points are clarified in the final manuscript with detailed explanations.

---

### Official Review · Reviewer_1LRA · 2025-07-01

**Clarity:** 4
**Significance:** 3
**Originality:** 4
**Rating:** 5
**Confidence:** 4

**Summary:**

The paper targets on the problem of dynamic view synthesis from a monocular video. Unlike previous works doing per-scene optimization or training a multi-view diffusion model, this paper proposes a new solution paradigm which is to construct the proper noise initialization, such that an off-the-shell video model could directly generate the view synthesis results. The paper first re-project the input video into a novel view video with holes using the depth prediction. Then it proposes a dedicated method to create a noise latent as the input to the diffusion model. In its core, the paper first analyze why a naive DDIM inversion won't produce satisfactory result due to zero terminal SNR, then proposes a recursive approach to craft this noise latent using the combination of DDIM and VAE, with the help of AdaIN to rescale the latent. The experiments are carefully designed which reveals the issue of the naive DDIM inversion as well as the effectiveness of the proposed method by comparing with existing works.

**Questions:**

See [Major Weakness; Minor Weakness / Questions] in the above section.

**Ethical Concerns:**

["NO or VERY MINOR ethics concerns only"]

**Final Justification:**

I have no major concerns about this paper. All my minor concerns have been well addressed. The proposed solution is a very refreshing perspective to solve the dynamic NVS problem so I recommend for acceptance.

**Limitations:**

Yes

**Paper Formatting Concerns:**

None.

**Quality:**

3

**Strengths And Weaknesses:**

Strengths:

1/ The idea is very interesting. As a researcher working on novel view synthesis via per-scene optimization, the proposed solution paradigm of finding the initial noise in an off-the-shell video model is very new to me.

2/ The solution is intuitively reasonable, elegant and without the need of massive model training. The motivation of well presented -- the proposed K-RNR emerge from the fact that naive SSIM inversion deviates from the training distribution so we need to stay within the distribution while preserve more data information in the initialization.

3/ Experiments are carefully designed to verify that the defects in the native solutions.

4/ Massive results in the supplemental materials serves as proof that the proposed method works as expected comparing to the existing approaches.

Major Weakness:

[Please correct me if I'm wrong. ] The only potential major weakness of the method is that it seems like the most important hyper-parameter, K, needs to be hand tweaked for different input videos: From the suppl. Videos it seems the elephant video is best recovered with K = 3, the duck one is best recovered with K=2, the monkey case seems to work the best with K=6 and the book-reading case needs K = 20. Any intuition picking K under than try-and-error?

Minor Weakness / Questions:

Does it only work for foreground-background videos? Have author tried on more complex videos such as a crowd of people dancing? More over does it work with larger movement in the camera trajectory? Do you have to tweak K for different trajectory of the same video?

From the suppl. videos, it seems TrajectoryCrafter works decently well most of the time. Hard to say the proposed method is definitely better than TrajectoryCrafter.

How to choose the intermediate index in AdaIN?

---

> ### Author Rebuttal · Authors · 2025-07-27
>
> ## How to choose $k$?
> **Q: From the suppl. Videos it seems the elephant video is best recovered with K = 3, the duck one is best recovered with K=2, the monkey case seems to work the best with K=6 and the book-reading case needs K = 20.**
> * We are thankful to reviewer for raising this practical concern. We provide additional details about our parameter selection process. Also, in `Appendix F.2: Discrete K-order Ablations section`, we conducted the quantitative experiments regarding different $k$ values.
> * **Q: Do you have to tweak K for different trajectory of the same video?** We fixed $k=10$ and $\delta=3$ in all quantitative experiments which yields the best results.
> * **Q: Any intuition picking $k$?** We obtained the best results when we set $k=10$ and $k=6$. Note that we don't tweak the $k$ value per video-camera pair. We also want to clarify an important point:
>     * In the **book reading** example, we presented video results for $k = 20$. This choice was not made because $k = 20$ is optimal, but rather because it represents a relatively high value of $k$. Our goal in that experiment is to highlight the effectiveness of our `adaptive normalization` extension of K-RNR when $k$ is high, which is why we chose to demonstrate the experiment at a higher $k$ setting.
>     * In the **monkey** example, we wanted to demonstrate the K-RNR's effect on rendered videos with increasing $k$ values. The logic behind that experiment is demonstrating readers the `evolution of videos` with different $k$ settings. As stated earlier, $k=6$ generates plausible results.
>     * In the **elephant** and **duck** examples, we aimed to demonstrate the effectiveness of K-RNR in source video reconstruction when there is no occlusion hence, no SLM involved.  We reported results using small values of $k$: [$k = 2$, $k = 3$, $k = 4$], to show that $k = 3$ is sufficient for direct video reconstruction. We will elaborate our parameter selection process in more detail in camera ready.
>
>
> ---
>
> ## How to choose $\delta$?
> **Q: How to choose the intermediate index $\delta$ in AdaIN?**
>
> *  In `Appendix F.3 Adaptive Reference Latent Index Ablations`, we conducted the quantitative experiments regarding different  values (in the table, the rows has typo where $k$ =... needs to be  $\delta$ = ...). In that experiment, we report PSNR, SSIM, LPIPS results. As a result of this experiment, we see that the best PSNR, SSIM and LPIPS scores are obtained when $\delta=3$. As a result of this **experimental validation**, in all of our experiments in the paper and supplementary videos, we use $\delta= 3$.
>
> ---
>
> ## Does it only work for foreground-background videos?
>
> **Q: Does it only work for foreground-background videos?**
>
> * Our work doesn't only work for foreground-background (such as `cat`, `monkey`, `book reading` examples) but also works in the presence of complex motions. Indeed, in the supplementary videos with captions `A police officer in uniform, accompanied by two men in mid-20th century overcoats and hats` and  `Two men in mid-20th century formal attire, including overcoats and fedoras, are on a boat` examples we shared two types of very complex motions:
>     * **A police officer in uniform, accompanied by two men in mid-20th century overcoats and hats |** Multiple people are walking in the original scene, moreover all of them performing diverse and very local hand gestures. We can see how K-RNR is preserving even the local hand gesture motions.
>     * **Two men in mid-20th century formal attire, including overcoats and fedoras, are on a boat |** This example is also very complex one as Di Caprio's hand movement is in slow motion and way too slower than the camera motion.
>     * Moreover, to show the effectiveness of our method in the presence of motion, different than baselines, we also shared a separate results on DAVIS dataset in the main paper.
>
> **Q: Have author tried on more complex videos such as a crowd of people dancing?**
> * Yes, we have evaluated our method on complex dancing scenes involving multiple people as requested. Specifically, we ran our model on such sequences and observed that our proposed method performs robustly even in the presence of dense, coordinated motion. Due to NeurIPS guidelines, we are not allowed to upload visuals, however these results will be added to the final manuscript.
> * We also note that our method is especially effective in cases where the source video has relatively low camera motion. In such scenarios, the video depth model (`DepthCrafter`) produces more accurate depth estimates, which in turn leads to minimal error in the 3D point cloud warping.
> * We will add more crowded, high complex motion scenes to our supplementary videos in the revised version. Specifically we will put the following complex dance scenes:
>     * The *La La Land* movie sunset dance scene (used by `ReCamMaster`), featuring a woman in a yellow dress and a man in a white shirt with a black tie.
>     * The tango scene from *Scent of a Woman* movie (used by `ReCamMaster`), set in front of a white piano, featuring a woman in a black backless dress and a man in a grey suit.
>
> ---
>
> ## TrajectoryCrafter vs Our Work
> **Q: From the suppl. videos, it seems TrajectoryCrafter works decently well most of the time. Hard to say the proposed method is definitely better than TrajectoryCrafter.**
>
> * Indeed, as discussed in `Appendix G. Discussion on Quantitative Results`, we elaborate our observations on TrajectoryCrafter among other methods. In particular, among the CogVideoX-based approaches, `Diffusion as Shader` struggles to maintain action fidelity, often generating semantically coherent frames that fail to reflect the intended motion trajectory. `TrajectoryCrafter` achieves a **stronger balance** between action fidelity and identity preservation; however, we note that identity consistency tends to degrade toward the latter segments of the video.
>     * This situation can be seen in 5th comparison video where old man is lying down on a bed.
>     * We also observed empirically that when it comes to synthesizing new hand movements, our work is more plausible than TrajectoryCrafter. This situation can be seen in the 1st and 4th comparison videos.
> * Moreover, it is important to note that TrajectoryCrafter requires training a custom `Ref-DiT` module built on top of CogVideoX. This module introduces Cross-Attention between view tokens and reference video tokens, and is inserted before every DiT block in CogVideoX.
> * In contrast, our method offers a significant efficiency advantage. The proposed K-RNR + SLM solution adds **less than one second** of additional latency to CogVideoX's own inference latency, making it far more lightweight.

---

> > ### Comment · Reviewer_1LRA · 2025-08-04
> >
> > Thanks for authors response, which has resolved all of my questions. It would be nice if the choice of k and delta could have some logic reason to back them up (to avoid hyper-parameter tuning in practice) but if not I think it's also not a big deal since the proposed method runs decently fast.
> >
> > I also appreciate authors' effort on promising more complex results in the revised version. It is unfortunate that we can't see the visuals per Neurips policy but I'm looking forward to seeing it in the final version.

---

> > > ### Author Response · Authors · 2025-08-04
> > >
> > > Dear Reviewer,
> > >
> > > We would like to thank you for your thoughtful follow-up. We’re glad our responses resolved all of your questions. We appreciate your suggestion on justifying the choice of $k$ and $\delta$, and we’ll aim to clarify this in the final version with our extended and more complex results.
> > >
> > > Best regards, Authors

---

### Official Review · Reviewer_Xvr7 · 2025-07-03

**Clarity:** 2
**Significance:** 2
**Originality:** 3
**Rating:** 4
**Confidence:** 3

**Summary:**

The paper proposes a training-free framework for dynamic view synthesis from a single monocular video. The task is formulated as an inverse problem to structure initial noises for a pre-trained video diffusion model. The paper poses a noise representation for input reconstruction and a feature modulation technique to handle newly visible regions. Experiments show better performance than prior works.

**Questions:**

* SLM samples latent patches from visible background regions to fill in newly revealed areas. This implicitly assumes the background is static or, at the very least, texturally homogeneous. Has the authors attempted to extend the framework to more general settings, possibly by incorporating more priors from video models?
* How much is the SNR collapse problem specific to CogVideoX model (or models adoptting zero terminal SNR schedules)? Does the proposed method applied to other video diffusion model backbones, e.g., Wan2.1 [1]?

[1] https://github.com/Wan-Video/Wan2.1

**Ethical Concerns:**

["NO or VERY MINOR ethics concerns only"]

**Final Justification:**

My concerns are addressed with the additional clarifications from the authors during the rebuttal. I found this work to achieve good empirical performance, though with some missing details, e.g., how positional embeddings are handled during resampling. Including these details will help improve the paper's clarity.

**Limitations:**

Yes.

**Quality:**

3

**Strengths And Weaknesses:**

Strengths
* The paper frames dynamic view synthesis as a noise initialization problem for a frozen video generation model. This enables development of training-free methods.
* Two techniques, K-RNR and SLM, are proposed to address two challenges in this task formulation, namely preserving fidelity to inputs and synthesizing unseen pixels. This is clearly presented and well motivated.

Weakness
* SLM samples from existing background regions and has inherent limitations. It may produce repetitive or semantically inconsistent results for in-the-wild videos, e.g., videos with cluttered backgrounds, or large camera movements (the latter is arguably discussed in the limitation).
* Looking at equation (2), the variance of $\epsilon^{(k)}$ is $(1 - \sqrt{1 - \bar{\alpha}_k}^k) \text{Var}(\epsilon^\text{inv})$, which should not explode (but should rather attenuate) with increasing $k$. More theoretical analysis and justifications on the empirical observation in Fig. 4 will strengthen the paper.

---

> ### Author Rebuttal · Authors · 2025-07-27
>
> ## Proof | Exploding Variance
> **Q: Looking at the Eq 2 the variance should not explode. More theoretical analysis and justification on the empirical observation in Fig. 4 will strengthen the paper.**
>
> We appreciate this insightful question. We show that indeed K-RNR variance is exploding with the increasing $k$ orders given a source video $\mathbf{V}$. We will focus on Eq 2 to derive variance for $\epsilon^{(k)}$. For your convenience, we rewrite Eq 2 below:
>
> $$
> \epsilon^{(k)} =  \sqrt{\bar{a}_t}\frac{1 - (\sqrt{1 - \bar{a}_t})^{k}}{1 - \sqrt{1 - \bar{a}_t}}x_0 + (\sqrt{1 - \bar{a}_t})^k\epsilon^{inv}  \tag{2}
> $$
>
> Then the variance of $\epsilon^{(k)}$ given the source video $\mathbf{V}$ is:
> $$
> \text{VAR}[\epsilon^{(k)} | \mathbf{V}] =  \text{VAR}[\sqrt{\bar{a}_t}\frac{1 - (\sqrt{1 - \bar{a}_t})^{k}}{1 - \sqrt{1 - \bar{a}_t}}x_0 | \mathbf{V}] + \text{VAR}[(\sqrt{1 - \bar{a}_t})^k\epsilon^{inv} | \mathbf{V}]  \tag{3}
> $$
>
> $$
> \quad \quad \quad \quad  =  \left[\sqrt{\bar{a}_t}\frac{1 - (\sqrt{1 - \bar{a}_t})^{k}}{1 - \sqrt{1 - \bar{a}_t}}\right]^2\text{VAR}[x_0 | \mathbf{V}] + (\sqrt{1 - \bar{a}_t})^{2k}\text{VAR}[\epsilon^{inv} | \mathbf{V}]  \tag{4}
> $$
>
> Here it is important to remind that given the source video $\mathbf{V}$, $\epsilon^{inv}$ is deterministic DDIM inverted latent where stochasticity parameter $\sigma$ is set to $0$ in the inversion process. Hence, $\epsilon^{inv}$ is not a random variable and $\text{VAR}[\epsilon^{inv}] = 0$. On the other hand $x_0$ is a random variable since it is sampled from VAE using reparameterization trick with non-zero variance.
>
> **Note:** While the DDIM inversion is deterministic, mapping the source video $\mathbf{V}$ into the VAE latent space involves sampling. To eliminate any stochasticity in this step, we use deterministic `mode sampling`, which returns the mean of the distribution with $0$ variance.
>
> Then the variance of $\epsilon^{(k)}$ reduces to:
> $$
> \text{VAR}[\epsilon^{(k)} | \mathbf{V}] =  \left[\sqrt{\bar{a}_t}\frac{1 - (\sqrt{1 - \bar{a}_t})^{k}}{1 - \sqrt{1 - \bar{a}_t}}\right]^2\text{VAR}[x_0 | \mathbf{V}]  =  \left(\frac{\sqrt{\bar{a}_t}}{1 - \sqrt{1 - \bar{a}_t}}\right)^2\left[1 - (\sqrt{1 - \bar{a}_t})^{k}\right]^2\text{VAR}[x_0 | \mathbf{V}]  \tag{5}
> $$
>
> Given that $\bar{a}_t \in (0, 1)$, we can inspect both extremes for the value of $k$.
>
> **When $k=1$:**
> $$
> \text{VAR}[\epsilon^{(1)} | \mathbf{V}] = (\sqrt{\bar{a}_t})^2\text{VAR}[x_0 | \mathbf{V}] \tag{6}
> $$
>
> **As $k \rightarrow \infty$:**
> $$
> \text{VAR}[\epsilon^{(\infty)} | \mathbf{V}] =  \left(\frac{\sqrt{\bar{a}_t}}{1 - \sqrt{1 - \bar{a}_t}}\right)^2\text{VAR}[x_0 | \mathbf{V}] = \left(\frac{1}{1 - \sqrt{1 - \bar{a}_t}}\right)^2(\sqrt{\bar{a}_t})^2\text{VAR}[x_0 | \mathbf{V}]  \tag{7}
> $$
>
> The quantities $\text{VAR}[\epsilon^{(1)} | \mathbf{V}]$ and $\text{VAR}[\epsilon^{(\infty)} | \mathbf{V}]$ differ by the multiplicative constant $\left(\frac{1}{1 - \sqrt{1 - \bar{a}_t}}\right)^2$ which is a very high number as $\bar{a}_t \in (0, 1)$. To be concrete, in our experiments $\bar{a}_t=0.0012$. As Eq 5 is monotone increasing with respect to $k$ and we show that as limit $k \rightarrow \infty$ the $\text{VAR}[\epsilon^{(\infty)} | \mathbf{V}] >> \text{VAR}[\epsilon^{(1)} | \mathbf{V}]$, **variance is exploding with increasing $k$.**
>
> ---
>
> ## Validity of SLM | How does SLM produce diverse inpaintings?
> **Q: Given that SLM resamples from visible regions to fill occluded regions, it can produces repetitive or cluttered patterns in the occluded regions./SLM assumes the background is static or, at the very least, texturally homogeneous.**
>
> * Although it may initially appear that the inpainting process simply reuses existing features potentially leading the SLM to struggle with repetitive or cluttered backgrounds, we kindly show that **this is not actually the case**.
>
> * The reason lies in the use of `positional embeddings`. Although existing features are reused for inpainting occluded regions, they are assigned different positional embeddings after the patchification process.
>
> * Specifically, **CogVideoX** adopts **3D Rotary Positional Embedding (3D RoPE)** for its relative positional encoding. In this framework, each token embedding is rotated by an angle proportional to its position. Concretely, an embedding of dimension $D$ is split into $\frac{D}{2}$ two‑dimensional chunks. Each chunk is rotated at a different angular frequency. Lower‑dimension chunks rotate more rapidly, encoding fine‑grained positional detail, whereas higher‑dimension chunks rotate more slowly and thus retain more semantic information. As a result, the higher dimensions are less affected by the token’s position and preserve more of its semantic content [1], hence, SLM becomes a context-aware sampling mechanism.
>
> * To showcase SLM's diversity, we present a comprehensive supplementary material with 100+ diverse results. Please view Supplementary/website.html
>
> ```
> [1] Barbero et al. "Round and round we go! what makes rotary positional encodings useful?" (2024). arXiv preprint arXiv:2410.06205. | Section 6.1 TRUNCATING THE LOWEST FREQUENCIES OF ROPE | "Summary of the section" part.
> ```
>
> ---
>
> ## Proof |  Generalizability of the Zero-Terminal SNR Collapse
> **Q: How much is the SNR collapse problem specific to CogVideoX model (or models adopting zero terminal SNR schedules)?**
>
> Thank you for this insightful technical question. In this section, we provide a formal proof that the identified Zero Terminal SNR Collapse is an architecture-agnostic phenomenon that arises solely due to the presence of zero terminal SNR.
>
> Our proof is based on the Maximum Entropy Principle as described in [1], specifically in **Theorem 5.6**.
>
> In Section 4.1, we demonstrated that given two source video latents $x_0$ and $y_0$, $\Phi_T(x_0, \epsilon) = \Phi_T(y_0, \epsilon) = \epsilon$ in the zero-terminal SNR setting which **breaks the injectivity**.
>
> Furthermore, it is known that the entropy of a random variable under deterministic transformation is always less than or equal to the original entropy:
>
> $$
> \mathcal{H}(\varepsilon_{\text{inv}}) = \mathcal{H}(\Phi^{-1}(x_0)) \leq \mathcal{H}(x_0),
> $$
>
> with equality only happens when $\Phi^{-1}$ is one-to-one, which we showed **does not hold true**. Hence in our case:
>
> $$
> \mathcal{H}(\Phi^{-1}(x_0)) = \mathcal{H}(\varepsilon_{\text{inv}})  < \mathcal{H}(x_0) \tag{1}
> $$
>
> Following the **Maximum Entropy Principle**, we begin by establishing an upper bound on the entropy. Our proof proceeds in two steps:
>
> 1. First, we will derive this upper bound for a general random variable $Z$.
> 2. We then apply the resulting inequality to the DDIM-inverted latent $\varepsilon_{\text{inv}}$ and the standard normal variable $\varepsilon$ (which represents the target terminal latent in the zero-terminal SNR setting).
>
> This comparison ultimately will show that the norm of $\varepsilon_{\text{inv}}$ is strictly less than that of $\varepsilon$, hence we are obtaining the norm deviation graph in **Figure 5**.
>
> The Maximum Entropy Principle gives the upper bound on the entropy. For a random variable $Z$ and any distribution with covariance $\Sigma$:
>
> $$
> h(Z) \leq \frac{1}{2} \log\left((2\pi e)^d \det \Sigma \right)
> $$
>
> Applying this Maximum Entropy Principle inequality to Eq. (1) above gives:
>
> $$
> \frac{1}{2} \log\left((2\pi e)^d \det \text{Cov}[\epsilon_{inv}] \right) < \frac{1}{2} \log\left((2\pi e)^d \det \text{Cov}[x_0] \right)
> $$
>
> or equivalently:
>
> $$
> \det\left(\text{Cov}[\epsilon_{inv}]\right) < \det\left(\text{Cov}[x_0]\right) \tag{2}
> $$
>
> Reminder: Given a source video $\mathbf{V}$, $x_0 = \text{VAE}[\mathbf{V}]$ which is trained to approximate multivariate Gaussian in its $d$ dimensional latent space. Hence $x_0 \sim \mathcal{N}(0, I_d)$. Based on this, rewriting Eq 2 gives:
>
> $$
> \det\left(\text{Cov}[\epsilon_{inv}]\right) < \det\left(I_d\right) = 1
> $$
>
> Hence every eigenvalue of $\text{Cov}(\varepsilon_{\text{inv}})$ is strictly below one, and therefore:
>
> $$
> \text{tr}(\text{Cov}(\varepsilon_{\text{inv}})) < d \tag{3}
> $$
>
> Now, we have everything to demonstrate the **expected norm deviation** formally. To demonstrate it, we will use the second moment identity of a random variable $Z$ in $R^d$:
>
> $$
> \mathbb{E}\left|Z\right|^2 = \text{tr}(\text{Cov}[Z]) + \mathbb{E}\left|Z\right|^2
> $$
>
> When applied to $\varepsilon_{\text{inv}}$ using the inequality (3), we get:
>
> $$
> \mathbb{E} \left\| \varepsilon_{\text{inv}} \right\|^2 = \text{tr}(\text{Cov}(\varepsilon_{\text{inv}})) + \mathbb{E} \left\| \varepsilon_{\text{inv}} \right\|^2 < d + \mathbb{E} \left\| \varepsilon_{\text{inv}} \right\|^2.
> $$
>
> In practice, $\varepsilon_{\text{inv}}$ is approximately centered (the network is trained with zero-mean terminal noise), giving $\mathbb{E} \left\| \varepsilon_{\text{inv}} \right\|^2 \approx 0$. This approximation is also verified in `Figure 4 (b)` where for expected noise (standard normal) norm (showed with dashes) and DDIM norm in $k=1$ axis  is in the similar scales.
>
> Thus,
>
> $$
> \mathbb{E} \left\| \varepsilon_{\text{inv}} \right\|^2 < d = \mathbb{E} \left\| \mathcal{N}(0, I_d) \right\|^2 = \mathbb{E} \left\| \varepsilon\right\|^2,
> $$
>
> which proves the claim.
>
> ```
> [1] Conrad, K. (2004). Probability distributions and maximum entropy. Entropy, 6(452), 10.
> ```
>
> ---
>
> ## Proposed methodology on Wan 2.1
> **Q: Does the proposed method applied to other video diffusion model backbones, e.g., Wan2.1?**
>
> * Our method relies on an inversion mechanism that maps a video back to the base distribution, just as DDIM inversion does in CogVideoX’s diffusion framework. In contrast, Wan is based on a flow matching model without a discrete step based reverse process, so DDIM style inversion cannot be applied directly.
>
> * Although flow models can be invertible under certain continuous normalizing flow formulations, to the best of our knowledge, to date there is no established inversion sampling procedure available for Wan.

---

> > ### Comment · Reviewer_Xvr7 · 2025-08-05
> >
> > Thank you authors for the detailed response. I have a few follow-up questions:
> > 1. How does the proposed stochastic permutation operator interact with 3D RoPE? Are the positional embeddings intact, while only the pre-PE latent features are resampled?
> > 2. Flow-based video models also admit ODE-based inversion by flipping the sign of the predicted velocity when doing ODE integration, which amounts to a discretization of the inverse of the ODE simulation in the standard denoising / forward video sampling procedure. Could you please explain more why this makes flow-based models incompatible?

---

> > > ### Author Response · Authors · 2025-08-05
> > >
> > > **Q: Are the positional embeddings intact, while only the pre-PE latent features are resampled?**
> > > That's right. The positional embeddings intact and only the pre-positional embedding latent features are resampled by Stochastic Latent Modulation.
> > >
> > > ## Non-iterative Inversion setting
> > > **Q: Proposed method on Wan 2.1 (or Flow Matching models in general)**
> > >
> > > We sincerely thank the reviewer for directing us toward Flow Matching models as a way to evaluate the effectiveness of our methodology. This provided a valuable opportunity to demonstrate that our reconstruction framework, K-RNR, along with our dynamic view synthesis approach, is directly compatible with Wan 2.1 without requiring any modifications. Furthermore, in the section below, we illustrate how K-RNR enables us to **bypass traditional iterative inversion schemes**, offering a more efficient, non-iterative alternative.
> > >
> > > * Without changing any architectural details, we directly used `pipeline_wan_video2video.py` from HuggingFace.
> > > * By default, it uses `UniPCMultistepScheduler` as the noise scheduler
> > > * In `add_noise` function of the noise scheduler $\epsilon^{`} = \alpha_t x_0 + \sigma_t \epsilon$ operation is performed where $x_0$ is the VAE encoded latent and $\epsilon$ is sampled from standard normal dist. Furthermore, $\alpha_t + \sigma_t = 1$. From now on, we will use $\sigma_t = (1 - \alpha_t)$ parameterization.
> > >
> > > We pose the following question: **How effective is K-RNR when used without relying on any inversion process?**
> > >
> > > To do so, we set $\epsilon^{(1)} = \epsilon \sim N(0, I)$ and we followed our recursive noise representation formula:
> > >
> > > $$
> > > \text{K-RNR in Flow Matching |} \quad \epsilon^{(k)} = \alpha_t x_0 + (1 - \alpha_t) \epsilon^{(k-1)} \tag{1}
> > > $$
> > >
> > > When this recursion is solved, we obtain a closed form solution again in the form of:
> > > $$
> > > \epsilon^{(k)} = \left[\sum_{i=1}^k (1-\alpha_t)^{i-1}\alpha_tx_0\right] + (1 - \alpha_t)^k\epsilon \tag{2}
> > > $$
> > >
> > > Importantly, $x_0$ is sampled from Wan 3D-VAE using `argmax-sampling` which uses the mode = mean of the latent distribution. Hence, $\mathbb{E}[x_0] = x_0$ and $\text{VAR}[x_0] = 0$.
> > >
> > > Now let's analyze the statistics and behavior of Eq.2:
> > >
> > > $$
> > > \mathbb{E}[\epsilon^{(k)}] = \left[\sum_{i=1}^k (1-\alpha_t)^{i-1}\alpha_t\mathbb{E}[x_0]\right] = \left[\frac{1 - (1 - \alpha_t)^k}{\alpha_t}\right]\alpha_t\mathbb{E}[x_0] = \left[1 - (1 - \alpha_t)^k\right]\mathbb{E}[x_0]
> > > $$
> > >
> > > $$
> > > \text{VAR}[\epsilon^{(k)}] = (1 - \alpha_t)^{2k}
> > > $$
> > >
> > > For the default setting, $\alpha_t = 0.07$.
> > >
> > > **Behavior of the mean |** When $k=1$, $\mathbb{E}[\epsilon^{(1)}] = 0.07\mathbb{E}[x_0]$. As $k\rightarrow \infty$, $\mathbb{E}[\epsilon^{(\infty)}] \rightarrow \mathbb{E}[x_0]$ so it gets $\frac{1}{0.07}\times \approx 15\times$ larger, hence **exploding**.
> > >
> > >
> > > **Behavior of the variance |** Note that we didn't use inverted latent for the $\epsilon^{(1)}$ but directly set it as standard normal different than our paper setting. And it results in a completely opposite results when it comes to variance behavior.  As $k\rightarrow \infty$, $\text{VAR}[\epsilon^{(\infty)}] \rightarrow 0$ hence it is **vanishing**.
> > >
> > > ## Validation of the theoretical results
> > > The reviewer can easily verify the theoretical results we demonstrated above. Indeed, it requires 4 lines of additional code to see the above results. Instead of using `latents = self.scheduler.add_noise(init_latents, noise, timestep)` in the `prepare_latents` function the following must be used:
> > >
> > > ```python
> > > k=3 # can be experimented with 1 (default), 5, 10, 50, 100 to see the theoretical results in action
> > > for i in range(k):
> > >     noise = self.scheduler.add_noise(init_latents, noise, timestep)
> > > latents = noise
> > > ```
> > >
> > > In our experiments when $k=3$ it becomes obvious that K-RNR has a structure match effect and has high fidelity structure preserving. However, as the mean explodes and variance vanishes, the color scheme is not natural. Indeed, when $k$ is very high, we see complete saturated examples as theory also shows.
> > >
> > > To overcome this, we again applied our `adaptive formulation` with $\delta=1$ and we got very sharp results.
> > >
> > > ## Another adaptive parameterization
> > > As in this setup, $\epsilon$ is not a deterministic inverted latent but a random variable, we can apply different adaptive parameterizations than the one we used in our paper.
> > >
> > > One possible parameterization is as variance is vanishing, setting a higher variance for $\epsilon$ in the first place:
> > >
> > > $$
> > > \epsilon^{`} = \epsilon \times \frac{1 - \alpha_t}{(1- \alpha_t)^k} = \epsilon \times \frac{1}{(1- \alpha_t)^{k-1}}
> > > $$
> > >
> > > And as the mean is exploding setting lower $\mathbb{E}[x_0]$ in the first place:
> > >
> > > $$
> > > x_0^{`} = x_0 \times \frac{\alpha_t}{1 - (1 - \alpha_t)^k}
> > > $$
> > >
> > > **Note:** the constants discussed above are chosen in a scale preserving manner. It can be proven that when above parametrization is used, the scale of mean and variance is the same as when $k=1$.
> > >
> > > ---
> > >
> > > We will make sure to add our findings for Wan 2.1 to the camera-ready version.

---

> > > > ### Comment · Reviewer_Xvr7 · 2025-08-08
> > > >
> > > > I thank the authors for the additional clarifications. My concerns are addressed and I will keep my positive score.

---

### Decision · Program_Chairs · 2025-09-17

**Decision:**

Accept (poster)

**Comment:**

This paper proposes a novel, training-free approach to dynamic novel view synthesis, offering a refreshing and promising perspective on a challenging problem. The initial reviews were mixed, with reviewers raising concerns about specific implementation details, clarity, and certain technical aspects. However, the authors provided a thorough and thoughtful rebuttal, effectively addressing all raised concerns through detailed clarifications and additional explanations.

Multiple reviewers acknowledged that their concerns were fully resolved. Reviewers broadly agreed that the method demonstrates strong empirical performance and makes a valuable contribution to the community. The Area Chair concurs with the reviewers’ final assessments and recommends acceptance. The authors are encouraged to integrate the clarifications and improvements discussed during the review process into the camera-ready version.